psychology/statistics

colour, emotion, multivariate pattern classification, machine learning, cultural specificity, Geneva Emotion Wheel (GEW)

**Author for correspondence:**
Daniel Oberfeld
e-mail: oberfeld@uni-mainz.de

# A machine learning approach to quantify the specificity of colour–emotion associations and their cultural differences

Domicele Jonauskaite[1], Jörg Wicker[2],
Christine Mohr[1], Nele Dael[1,3], Jelena Havelka[4],
Marietta Papadatou-Pastou[5], Meng Zhang[6]
and Daniel Oberfeld[7,8]

[1]Institute of Psychology, University of Lausanne, Lausanne, Switzerland
[2]School of Computer Science, University of Auckland, Auckland, New Zealand
[3]Department of Organizational Behavior, University of Lausanne, Lausanne, Switzerland
[4]School of Psychology, University of Leeds, Leeds, UK
[5]School of Education, National and Kapodistrian University of Athens, Athens, Greece
[6]Department of Psychology and Behavioral Sciences, Zhejiang University, Hangzhou, People's Republic of China
[7]Institute of Psychology, Johannes Gutenberg-Universität, Mainz, Germany
[8]Laboratoire ICube UMR7357 Université de Strasbourg, France

 
DJ, 0000-0002-7513-9766; JW, 0000-0003-0533-3368
DO, 0000-0002-6710-3309

The link between colour and emotion and its possible similarity across cultures are questions that have not been fully resolved. Online, 711 participants from China, Germany, Greece and the UK associated 12 colour terms with 20 discrete emotion terms in their native languages. We propose a machine learning approach to quantify (a) the consistency and specificity of colour–emotion associations and (b) the degree to which they are country-specific, on the basis of the accuracy of a statistical classifier in (a) decoding the colour term evaluated on a given trial from the 20 ratings of colour–emotion associations and (b) predicting the country of origin from the 240 individual colour–emotion associations, respectively. The classifier accuracies were significantly above chance level, demonstrating that emotion associations are to some extent colour-specific and that colour–emotion associations are to some extent country-specific. A second measure of country-specificity, the in-group advantage of the colour-decoding accuracy, was detectable but relatively small (6.1%), indicating that colour–emotion associations are both universal and culture-specific. Our results show that machine learning is a promising tool when analysing complex datasets from emotion research.

# 1. Introduction

In the case of an intact visual system, humans experience an environment rich in diverse colours (e.g. [1]). Plants, animals and landscapes—all provide a large palette of colour experiences. We also frequently choose to add colour to ourselves (e.g. make-up, clothes) or surrounding objects (e.g. cars, interior spaces, decorative objects). These colour choices are rarely random, instead they tend to be based on our colour preferences [2], perceived colour appropriateness [3,4], and perceptual qualities that a colour could give to an interior space [5–7]. Artists use colour to represent properties of painted objects [8]. Colours seem to add perceptual and affective qualities to our visual environment.

Associations between colours and emotion have been studied for more than a century (e.g. [9–11]). If we take the example '*red*', numerous colour-related metaphors, like '*seeing red*' (i.e. being angry), exist in various languages including English, German and Chinese [12–15]. Complementary to linguistic metaphors, red has been reported to be associated with danger or anger in the USA [15,16], the UK [17,18], France [19] and China [20]. In addition to negative connotations, red seems to be also associated with positive emotions in Spain [13], the UK [21] and China [20]. As we can see from these examples, a single colour may carry diverse connotations which may or may not be the same for different countries.

However, the question of how *specific* and *consistent* colour–emotion associations are across different individuals has not been adequately answered. Over 60 years ago, Wexner [10] reported that red was often associated with 'exciting', but also with 'joyful', 'hostile' or 'powerful'. Findings like these argue against a one-to-one mapping between colour and discrete emotions. Moreover, such results do not indicate whether the four emotion associations reported by Wexner are colour-specific. Although it is possible that they only occur for red, it is also possible that the same associations are present for other colours such as blue or green. We also do not know how consistent colour–emotion associations are across participants. It might be the case that a person who associates red with excitation also associates it with joy and hostility. Alternatively, reported associations can differ between participants, with one participant associating red only with excitation, a second participant only with joy and a third participant only with 'hostile'.

In the current contribution, we propose a machine learning/statistical learning approach (multivariate pattern classification; [22,23]) to quantify the specificity and consistency of associations between colour terms and discrete emotions. In an online study (https://www2.unil.ch/onlinepsylab/colour/main.php), 711 participants from four different countries (China, Germany, Greece and the UK) rated the association of 12 colour terms with 20 discrete emotions and indicated the intensity of each associated emotion [24]. Thus, from each participant, we obtained ratings for 240 colour–emotion associations. We used this dataset to test for the consistency and specificity of colour–emotion associations. To implement this approach, we divided the response dataset into a training set and a test set. First, we trained a statistical classifier [23] using the colour–emotion associations from the training set. Then, for each individual set of association ratings of 20 emotions in the test set, we used the classifier to predict the corresponding colour term.

If emotion associations are highly colour-specific and consistent across participants, it should be possible to predict at high accuracy which colour term a participant had evaluated, given the 20 emotion ratings. In contrast, if the colour–emotion associations are idiosyncratic or do not differ systematically between colour terms, the classifier accuracy should be close to chance level. Thus, high accuracy of the classifier for the test set indicates consistent and colour-specific emotion associations, while low accuracy indicates emotion associations which are shared across several colour terms and/or are very heterogeneous between participants. For this reason, the accuracy of the classifier in decoding the colour rated on a given trial from the 20 colour–emotion associations provides a *quantitative* measure of the specificity and consistency of the colour–emotion associations.

With regard to the universality of colour–emotion associations, we have some knowledge of *cultural differences* in affective connotations of colour. In a seminal study, Adams & Osgood [25] analysed associations between colour terms and emotion dimensions of valence (positive–negative), arousal (high arousal–low arousal) and potency (dominant–submissive) in 23 countries. They found a high degree of cross-cultural similarity of these associations (e.g. black was always rated as negative, while white, blue and green were rated as positive). The authors concluded that there are strong universal tendencies in colour–emotion associations (for further similarities between cultures, see also [26–28]). In contrast, Hupka *et al.* [29] reported cultural differences. They tested to what extent anger, fear, envy and jealousy reminded participants of specific colour terms. Comparing results across five countries,

one prominent finding was that cross-cultural disagreement arose over colour associations with envy, because Germans linked envy much more frequently to yellow than any other nation (also see [30]).

To quantify the degree to which colour–emotion associations are *country-specific*, we propose to use the same machine learning rationale as for the colour-specificity question. If associations between colour and emotion are country-specific, then it should be possible to predict a participant's country of origin using his or her set of 240 colour–emotion association ratings. Just as for the specificity of the colour–emotion associations, the accuracy of the classifier in predicting the country of origin from the 240 individual colour–emotion associations provides a quantitative measure of the country-specificity of the colour–emotion associations. As a second measure of the country-specificity, we propose the *in-group advantage* of the colour-decoding accuracy (e.g. [31]). For instance, if there is a considerable difference between the colour–emotion associations in China and Germany, a classifier trained on Chinese data should show a considerably higher accuracy in predicting the evaluated colour term from a given set of 20 colour–emotion associations for participants from China, compared with participants from Germany.

To sum up, we extracted data from an online study which collected ratings of associations between 12 colour terms and 20 discrete emotion terms [24]. In the current paper, we present data from China, Germany, Greece and the UK. We propose that colour-specificity of emotion associations and country-specificity of colour–emotion associations can be measured using a multivariate pattern classification approach (*statistical learning/machine learning*), to answer the following three questions:

(1) Are colour–emotion associations systematic enough to predict from the response pattern which colour term was evaluated?
(2) Are cultural differences in colour–emotion associations systematic enough to decode a participant's country of origin from his or her response pattern?
(3) Is there an in-group advantage when predicting an evaluated colour term from the response pattern and if so, how large is it?

# 2. Methods

## 2.1. Participants

The data presented here have been extracted from an ongoing online survey [24]. We extracted and analysed data from four countries, China, Germany, Greece and the UK. We selected these four countries because we had convenient access to study participants, and could include two European countries that were studied relatively often in the colour–emotion literature (the UK and Germany; e.g. [29]), one Eastern culture (e.g. [32]), and one European country that has been studied less often (Greece) and also uses a non-Latin alphabet. We do not imply that these four countries are representative for all countries in the world. *A priori*, we aimed for a relatively large age range, i.e. we started data extraction and data analysis only when we had about 30 participants in each of three age groups (16–29 years, 30–49 years and 50+ years) for each of the four countries. Seven hundred and thirty-five participants from the four countries had completed the survey at the time of the current data extraction.

Participation was voluntary. The experiment was conducted in compliance with the ethical standards described in the Declaration of Helsinki. On the first page of the survey (https://www2.unil.ch/onlinepsylab/colour/main.php), we provided ethical information (i.e. participation is anonymous/strictly confidential, the responses will be used for research purposes only, participants can stop the survey at any time without any consequences). Participants were informed that they would express their consent to participate by clicking on the '*Let's go*' button. Some participants received partial course credit for participation, but most completed the survey on a purely voluntary basis with no direct compensation.

As the data came from an online survey, we applied relatively strict criteria to select valid responses. We included only participants who completed the survey in their native language, which coincided with an official language of their country of origin (e.g. German speakers originating from Germany, who filled in the survey in German). We excluded participants reporting potential problems with colour vision (i.e. who responded '*yes*', '*I don't know*' or '*do not want to answer*' to the question '*Do you have problems seeing colours?*'). Participants who were very quick (took less than 5 min on the main task) or very slow (took more than 60 min) in completing the survey were also excluded. Finally, we excluded participants who did not show minimal engagement with the task (i.e. spent less than 20 s on the first four colour terms). Applying these criteria, 711 participants entered the data analysis (table 1).

**Table 1.** Demographic data of the participants included in the data analysis, for each of the four countries.

|  | Germany | UK | Greece | China |
|---|---|---|---|---|
| female: $N$ | 172 | 87 | 220 | 101 |
| male: $N$ | 22 | 39 | 31 | 37 |
| age in years: mean (s.d.) | 33.3 (16.1) | 40.6 (13.5) | 30.4 (11.6) | 36.1 (18.3) |
| age range in years | 16–75 | 16–71 | 15–76 | 17–79 |
| total: $N$ | 195 | 126 | 252 | 138 |

The mean age differed significantly between the four countries; $F_{3,695} = 14.4$, $p < 0.001$, $\eta_p^2 = 0.059$ (table 1). *Post hoc* pairwise comparisons (*t*-tests) showed that, on average, Greek participants were significantly younger than British ($p < 0.001$, two-tailed; all Bonferroni corrected, reported *p*-values are multiplied by the number of tests) and Chinese ($p = 0.002$) participants. German participants were significantly younger than British participants ($p < 0.001$). The gender ratio also differed significantly between countries; $\chi^2(3) = 32.9$, $p < 0.001$ (table 1) with more female participants taking part in each country.

## 2.2. Colour terms

We used the 11 basic colour terms proposed by Berlin & Kay [33]: WHITE, BLACK, GREY, RED, YELLOW, GREEN, BLUE, ORANGE, PURPLE, PINK and BROWN. In addition, we included TURQUOISE because it is an emerging basic colour term in English [34] and an already existing colour term in several European languages [35]. Note that this selection of colour terms focuses on differences in hue. Nevertheless, some colour terms also describe variation in saturation (PINK versus RED) or brightness (BLACK, WHITE, GREY), and also effects of the brightness contrast between a coloured area and its surrounding (ORANGE versus BROWN, GREY versus BLACK or WHITE). We refer to cognitive colour categories using capital letters (e.g. RED). Please note that we used linguistic colour terms rather than, e.g., colour patches to assess semantic associations because it is impossible to control the colorimetric coordinates of colours in an online survey.

## 2.3. Emotion assessment: Geneva Emotion Wheel

We used the validated Geneva Emotion Wheel (GEW, v. 3.0; [36,37]) to assess emotion associations of each colour term with 20 discrete emotion terms such as 'love', 'hate' or 'compassion'. As shown in figure 1, the spatial position of the emotions in the GEW represents their relative proximity along two major appraisal dimensions: goal conduciveness (valence) and coping potential (control/power/dominance) [37,38]. Emotions appearing in the left quadrants have negative valence while in the right quadrants, the valence is positive, with the exception of compassion that seems neutral in valence [37]. The lower quadrants correspond to lower power and the upper quadrants correspond to higher power. A square and five circles of increasing size, distributed from the hub to the rim of the wheel, signify six degrees of intensity of the emotions. The square represents no emotion. The larger circles on the rim of the wheel represent stronger emotions.

We use numerical codes when referring to the six rating categories on the GEW. The rating category indicating no association between the colour term and a given emotion (i.e. the square on the GEW) was coded as 0. The size of the five circles indicates the intensity of an associated emotion, ranging from 1 (weak emotion) to 5 (strong emotion).

By default, the square (representing the absence of an association between the given colour term and a given emotion) was selected for each emotion when the participant started to provide emotion associations for a given colour term. In addition, the options *'No emotion'* and *'Different emotion'* appeared in the centre of the wheel. By clicking on *'No emotion'*, the participant was able to indicate that he or she did not endorse any association between the colour term and any of the 20 emotion terms. If participants clicked on *'Different emotion'*, a pop-up window appeared and they were invited to freely type the emotion or emotions they had in mind. We did not further analyse the emotions that were associated to *'Different emotion'* in the current study.

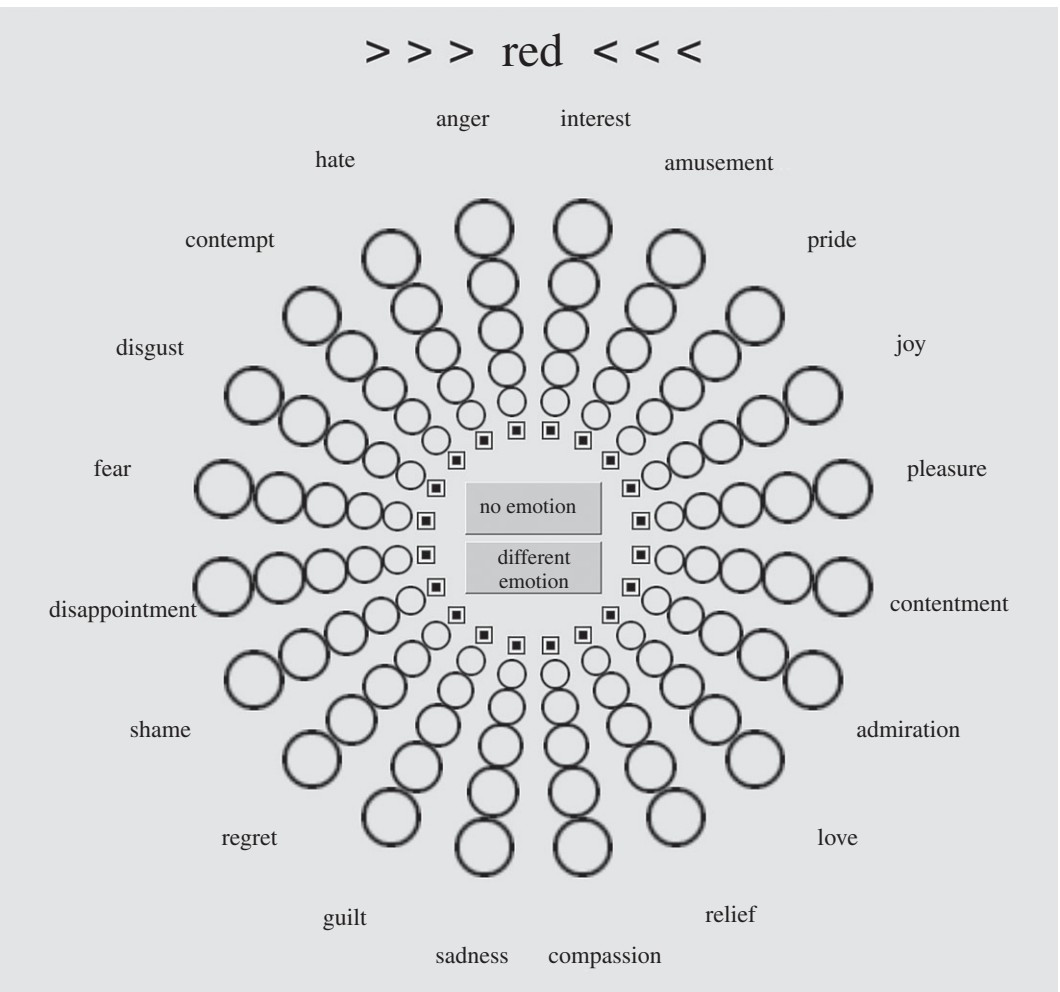

**Figure 1.** Participants used the Geneva Emotion Wheel [37] to rate the associations between 20 discrete emotions and a colour term. They selected the square to indicate that they perceived no association between the colour term shown at the top (RED in this example) and the given emotion. The largest circle represents the highest possible intensity of an emotion associated with the colour term. See table 2 for colour and emotion terms in German, Greek and simplified Mandarin Chinese.

English, German and Traditional Mandarin Chinese versions of the GEW are available from the Swiss Centre for Affective Sciences (http://www.affective-sciences.org/gew). Our collaborators and co-authors converted/translated the GEW into simplified Mandarin Chinese (author M.Z.) and Greek (see also Acknowledgements). To ensure that the meaning of the translated emotion terms was as close as possible to the meaning of the original emotion terms (in English), we used a back-translation technique. One translator (a bilingual person in the target and the reference language) translated the emotion terms into the target language. Then, the second translator (again a bilingual person in the target and the reference language) translated the emotion terms from the target to the reference language without having seen the original equivalents in the reference language (table 2). Next, the reference and back-translated versions were compared. Possible discrepancies were resolved through discussion and consultation of dictionaries. The main instruction text was originally written in English. Some of the current authors translated the text into their respective languages: German (D.O. and C.M.), Greek (M.P.-P.) and simplified Mandarin Chinese (M.Z.).

## 2.4. Procedure

On the first page of the survey, we provided information about ethical issues relevant to the participation in the study (see above). Participants expressed their consent to participate by clicking on the 'Let's go' button. The following two instruction pages explained the task and the use of the GEW. We then tested their task comprehension using a fictitious rating they had to correct according to the instruction: '*Peter thinks that beige strongly represents intense compassion, and believes that beige is also associated with mild relief. Accidentally, he has selected sadness and wants to correct his choice. Look at his response in the emotion wheel*

**Table 2.** Colour and emotion terms in English, German, Greek and simplified Mandarin Chinese used in this study.

| English | German | Greek | simplified Mandarin Chinese |
|---|---|---|---|
| black | Schwarz | Μαύρο | 黑色 |
| blue | Blau | Μπλέ | 蓝色 |
| brown | Braun | Καφέ | 棕色 |
| green | Grün | Πράσινο | 绿色 |
| grey | Grau | Γκρι | 灰色 |
| orange | Orange | Πορτοκαλί | 桔色 |
| pink | Rosa | Ροζ | 粉色 |
| purple | Lila | Μωβ | 紫色 |
| red | Rot | Κόκκινο | 红色 |
| turquoise | Türkis | Γαλάζιο | 青色 |
| white | Weiss | Λευκό | 白色 |
| yellow | Gelb | Κίτρινο | 黄色 |
| admiration | Bewunderung | Θαυμασμός | 赞赏 |
| amusement | Belustigung | Διασκέδαση | 欢愉 |
| anger | Ärger | Θυμός | 忿怒 |
| compassion | Mitgefühl | Συμπόνια | 同情 |
| contempt | Verachtung | Περιφρόνηση | 轻视 |
| contentment | Zufriedenheit | Ικανοποίηση | 满足 |
| disappointment | Enttäuschung | Απογοήτευση | 失望 |
| disgust | Ekel | Αηδία | 厌恶 |
| fear | Angst | Φόβος | 恐惧 |
| guilt | Schuld | Ενοχή | 内疚 |
| hate | Hass | Μίσος | 憎恨 |
| interest | Interesse | Ενδιαφέρον | 感兴趣 |
| joy | Freude | Χαρά | 欢乐 |
| love | Liebe | Αγάπη | 爱 |
| pleasure | Vergnügen | Ευχαρίστηση | 愉快 |
| pride | Stolz | Υπερηφάνεια | 自豪 |
| regret | Bereuen | Μετάνοια | 后悔 |
| relief | Erleichterung | Ανακούφιση | 如释重负 |
| sadness | Trauer | Θλίψη | 悲伤 |
| shame | Scham | Ντροπή | 羞愧 |

*below and try to correct it'*. Participants could only move to the next page and start the survey if they clicked on the square for sadness (no association), the largest circle for compassion (emotion intensity 5) and the middle circle for relief (emotion intensity 3). This allowed us to be sure they understood both how to use the GEW and the task.

Next, the actual task started. Participants saw 12 colour terms in their native language (table 2), appearing above the GEW one at a time in randomized order. They were asked to rate the intensity of the association between each of the 20 emotion terms contained in the GEW and the given colour term on the 6-point rating scale explained above. The participants were only allowed to proceed to the next colour term if they had assigned at least one rating, i.e. (1) associated at least one emotion with the given colour term (i.e. had clicked on at least one of the circles on the GEW), (2) had clicked on the '*No emotion*' response button, or (3) had specified an association with an emotion not contained in the GEW (i.e. clicked on the '*Different emotion*' button).

After evaluating the 12 colour terms, participants completed a demographic questionnaire on which they reported their age, gender, potential colour vision impairments, importance of colour in their life, country of origin, country of residence, native language and fluency in the language the colour–emotion survey was completed in. Participants could decline to answer the demographic questions. On the final page, participants were thanked and received results from a previous, related study in a graphic format. We provided an e-mail address in case participants had questions or comments regarding the study, or wished to be informed about this or other research studies. The entire survey took on average 13 min to complete.

## 2.5. Data analysis

We used statistical learning (multivariate pattern classification) to predict the colour term a participant evaluated on a given trial from his or her set of 20 ratings of emotion association. For classification, we selected the support vector machine approach (SVM; cf. [23,39]) with a radial basis function (RBF) kernel, and used error-correcting output codes (ECOC) for the multiclass classification [40]. To optimize the parameters of the SVM (complexity constant $C$ and the $\lambda$-parameter of the RBF kernel), we used a grid search or Bayesian optimization.

To evaluate the accuracy of the classifier, a 10-fold cross-validation (CV) was conducted (cf. [22]). To account for the repeated-measures structure of the data, the CV was based on participants. Thus, in a given iteration of the CV, approximately 9/10 of participants were randomly selected to be in the training set, and the resulting classifier was then applied to the data of the remaining approximately 1/10 of participants. As a consequence, we never tested the classifier on data coming from a participant who was included in the training set. This approach enabled us to get a realistic estimate of the consistency of the colour–emotion associations across participants and to avoid information leakage. We compared the performance of the classifier with the performance of the same method on 10 randomized datasets. The randomized datasets were generated by randomly permuting the class values (i.e. colour terms) of the dataset [41]. Therefore, we could decide whether the classification of the actual data provided a significantly higher performance than the classification based on the randomized data. The latter represents baseline performance.

A summary measure of the predictive power of a classifier is the area under the receiver operating characteristic (ROC) curve (AUC; e.g. [42]). This measure provides information about the degree to which the predicted country of origin is concordant with the actual country of origin. Areas of 0.5 and 1.0 correspond to performance at chance level and perfect performance of the classifier, respectively. AUC is not affected by response bias or by the prior probabilities of the classes.

We used similar methods as for the colour-classification to predict the country of origin of a participant from his or her set of 240 ratings of colour–emotion association. Because the sample sizes differed between our four countries (table 1), we used a uniform prior when training and evaluating the classifier, so that the results were not affected by the differing prior probabilities of the four classes (i.e. countries). The analyses were implemented in the Weka 3 data mining software (http://www.cs.waikato.ac.nz/ml/weka/; [43]) and in Matlab (function *fitcecoc*).

# 3. Results

## 3.1. Consistency and specificity of colour–emotion associations across countries

In the first set of analyses, we investigated whether the emotion associations are colour-specific. To this end, we analysed the data from all four countries together. We first present some descriptive analyses. Then, we introduce the machine learning approach using the accuracy of a statistical classifier in decoding the colour evaluated on a given trial from the 20 ratings of colour–emotion associations as a measure of the consistency and specificity of colour–emotion associations.

### 3.1.1. Intensity ratings of emotions associated with colour terms

The ratings of the intensity of emotion associations for the 12 colour terms are depicted in figure 2. Each panel represents a colour term. For each combination of a colour term and an emotion category, the different areas in the stacked bar represent the proportion of participants that chose a particular emotion intensity. For instance, participants might have responded that the given emotion was not associated with the given colour (rating category 0; grey area), or that the emotion was strongly associated with the colour (rating category 5; red area). Thus, within each colour, a larger non-grey

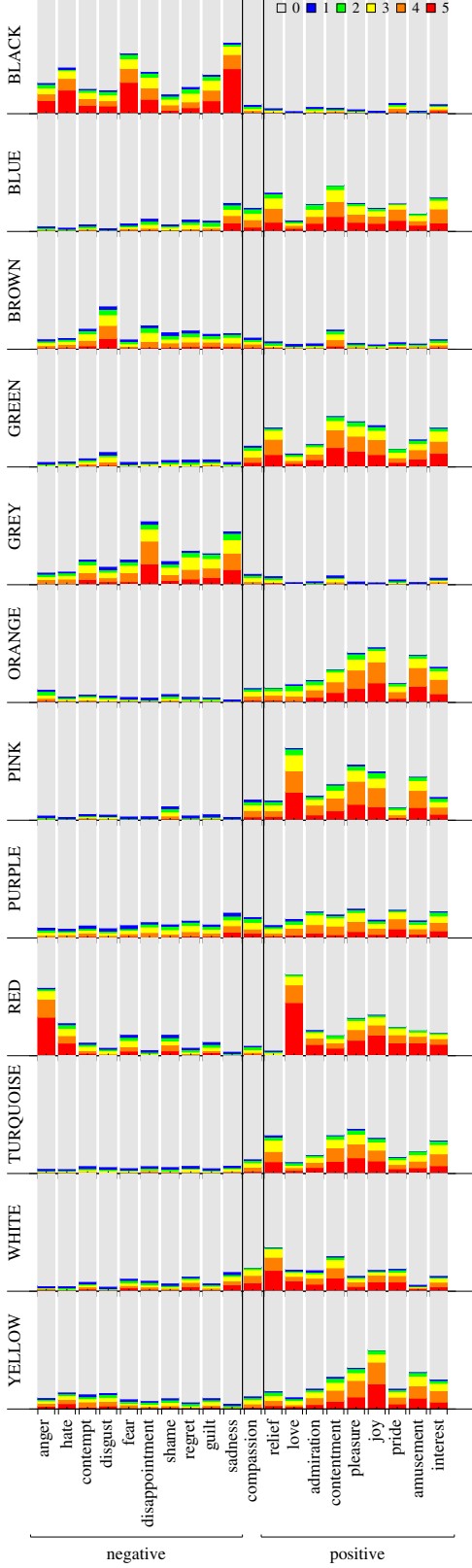

**Figure 2.** Stacked bar charts showing how often participants chose the six GEW intensity ratings (unweighted average proportions across countries). Each panel represents a colour term. On the horizontal axis, emotions on the left-hand side of 'compassion' have negative valence, emotions on the right-hand side of 'compassion' have positive valence and 'compassion' is neutral in valence [37]. The grey areas represent the average proportion of participants that selected rating category 0 (i.e. no colour–emotion association; squares on the GEW) for a given combination of colour term and emotion. The red areas represent the proportion of participants that selected rating category 5 (i.e. the strongest emotion intensity; largest circles on the GEW). The blue, green, yellow and orange areas represent the proportions of increasing intermediate ratings of emotion intensity (smallest to second largest circle on the GEW).

area indicates a higher proportion of emotion associations. Due to the different sample sizes in the four countries, we computed unweighted means by first computing the proportions of each rating category within each country, and then averaging the proportions across countries. Particularly strong colour–emotion associations were observed for RED, BLACK and PINK, and particularly weak associations were observed for BROWN and PURPLE. Across the 240 colour–emotion pairs, the strongest association was found between the emotion *love* and the colour term RED. Here, the bar chart shows that if this association was reported, it tended to be strong, that is, the rating category 5 on the GEW was most frequently selected. Similar patterns were observed for the associations RED–anger and BLACK–sadness. In contrast, participants indicating an association between YELLOW and pleasure frequently selected lower rating categories on the GEW, indicating a comparably weak, yet frequent, colour–emotion association.

Figure 2 shows that most colours were associated with either predominantly positive or predominantly negative emotions. For instance, BLACK was rarely associated with positive emotions, while PINK was rarely associated with negative emotions. A notable exception is RED, which was most frequently associated with love and other positive emotions, but also with anger, fear, hate or shame. For most colour terms, as for example BLUE, the colour was predominantly associated with either positive or negative emotions. However, a broader range of emotions within the corresponding positive or negative valence category was selected. It was only rarely the case that a discrete emotion had a unique, strong association with a colour term. The notable exception was RED–love, and to a lesser extent also RED–anger, BROWN–disgust, PINK–love and BLACK–sadness.

### 3.1.2. Colour classification based on the ratings of colour–emotion associations

The classifier (SVM) predicted the colour evaluated on a given trial on the basis of the 20 ratings of colour–emotion association. Across the 10 cross-validations, the area under the ROC curve (AUC) was 0.830 (38.7% correctly classified instances), which represents a moderately high classification accuracy. In comparison, for the randomly permuted datasets, the AUC was 0.52, which is significantly smaller ($p < 0.05$). Note that because there were 12 different colour terms, the guessing rate for the colour classification is 1/12 (8.33%), which is well below the 38.7% correctly classified instances. Thus, the observed classifier accuracy is higher than chance level, indicating that the patterns of the 20 emotion associations are moderately specific for the 12 colour terms.

Figure 3 shows that the classification accuracy (proportion of instances where the predicted colour term corresponded to the true colour term; i.e. true positive rate, or recall) was highest for BLACK and RED, followed by BROWN, PINK and GREY. For the remaining colour terms, the classification accuracy was considerably lower. Thus, BLACK, RED, BROWN, PINK and GREY elicited relatively specific emotion associations. The confusion matrix also shows that, for instance, ORANGE was frequently misclassified as YELLOW, indicating a relatively high similarity between the emotion associations for these two colour terms.

The moderately high level of classifier accuracy was obtained in 10-fold cross-validation, where the classifier is repeatedly trained on a random subset of the data (training set) and then the accuracy is measured for the subset of the data (test set) not included in the training set. For this reason, it is unlikely that the relatively good performance of the classifier could be due to overfitting. As mentioned above, the data from four countries analysed here are from a large-scale, cross-cultural survey on colour–emotion associations [24] that includes data from participants from 30 nations (located on all continents but Antarctica; the data can be accessed at https://forsbase.unil.ch/datasets/dataset-public-detail/15126/1474/). We applied the colour classifier trained on the data from China, Greece, Germany and the UK to the data from the remaining 26 countries in the survey. This dataset from the 26 countries contained ratings of colour–emotion association from 3222 participants. The classification accuracy was 30.4% and the AUC was 0.704. While these values are somewhat lower than for the four-countries dataset (see above), they clearly demonstrate classifier performance well above chance level. Put differently, we successfully tested the machine learning algorithm on an independent sample.

## 3.2. Cultural differences

In the second set of analyses, we investigated the degree of cultural differences in colour–emotion associations. We first present some descriptive analyses. Next, we quantify the country-specificity of colour–emotion associations, first by the accuracy of a classifier decoding a participant's country of

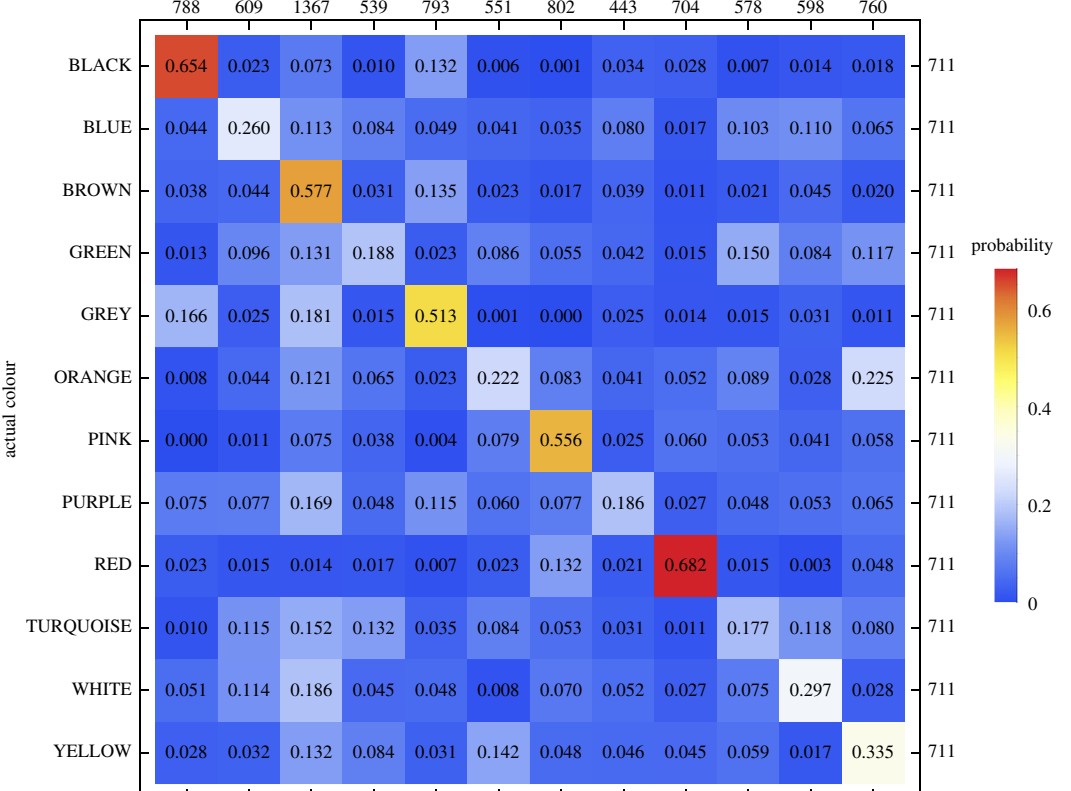

**Figure 3.** Confusion matrix for the prediction of the colour term evaluated on a given trial based on the set of 20 colour–emotion association ratings provided by each of the 711 participants (classification algorithm: optimized support vector machines). Rows represent the actual and columns the predicted colour terms, respectively. The number in each cell represents the proportion of trials presenting the colour term specified on the vertical axis (row) that were classified as corresponding to the colour term specified on the horizontal axis (column). Thus, proportions on the main diagonal represent true positive rate or recall. The numbers on the right-hand side of the frame represent the absolute frequency of the actually presented colours. The numbers on the upper side of the frame represent the absolute frequency of predicted colours.

origin from his or her 240 colour–emotion association ratings, and then by the within-country advantage when predicting the evaluated colour from the 20 ratings of emotion association.

### 3.2.1. Per-country intensity ratings of emotions associated with colour terms

Figures 4–6 show the colour–emotion association ratings separately for each of the four countries (i.e. China, Germany, Greece and the UK). Although the response patterns are relatively similar across countries, this descriptive analysis reveals some cultural differences. To give some examples, the association between BROWN and disgust was stronger in Germany than in the remaining countries and almost non-existent in China (figure 4). Participants from Greece were the only group that strongly expressed an association between PURPLE and sadness (figure 5). WHITE was more frequently associated with negative emotions in China compared with the three other countries (figure 6). Finally, YELLOW was primarily associated with positive emotions, such as joy, in all countries apart from Greece where it was frequently associated with negative emotions (figure 6).

### 3.2.2. Country classification

To quantify the degree to which colour–emotion associations are *country-specific*, we applied the same machine learning rationale as above. If the colour–emotion associations are country-specific, then it should be possible to predict from which country a given participant originates from his or her set of 240 colour–emotion associations. Across 10 cross-validations, the weighted average area under the

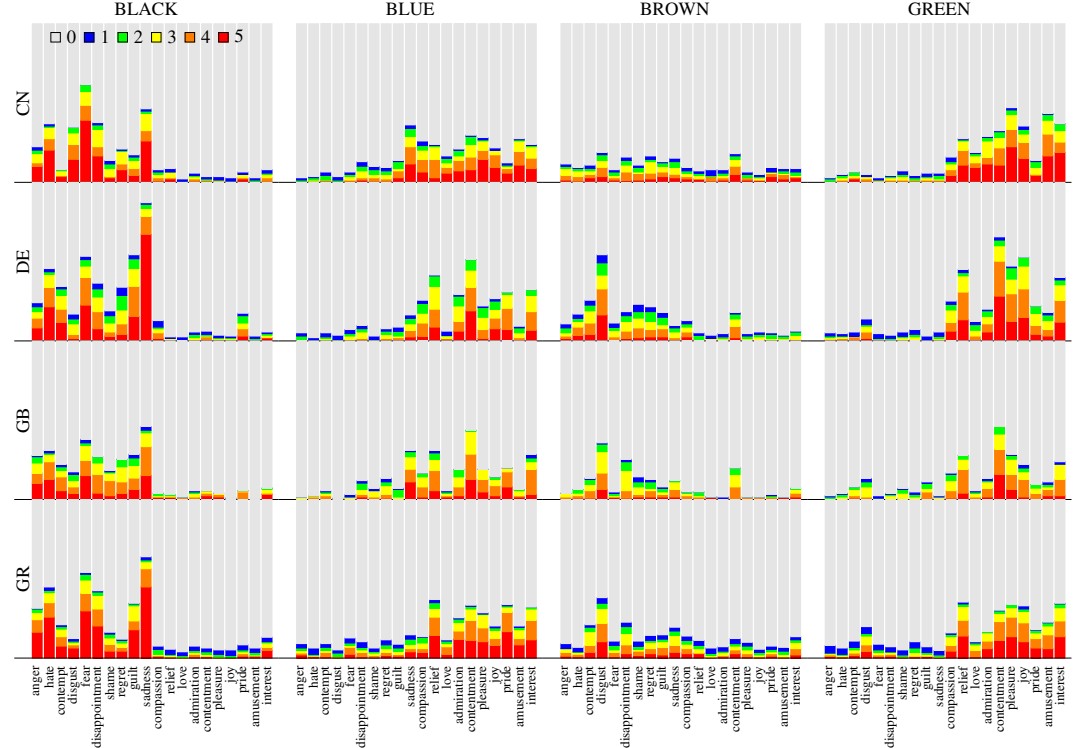

**Figure 4.** Colour terms BLACK, BLUE, BROWN and GREEN: stacked bar charts showing proportions of the six GEW rating categories representing the intensity of the associated emotions, separately for the four countries. Same colour code as in figure 2. Rows represent countries (CN: China, DE: Germany, GB: United Kingdom, GR: Greece) and columns represent colour terms.

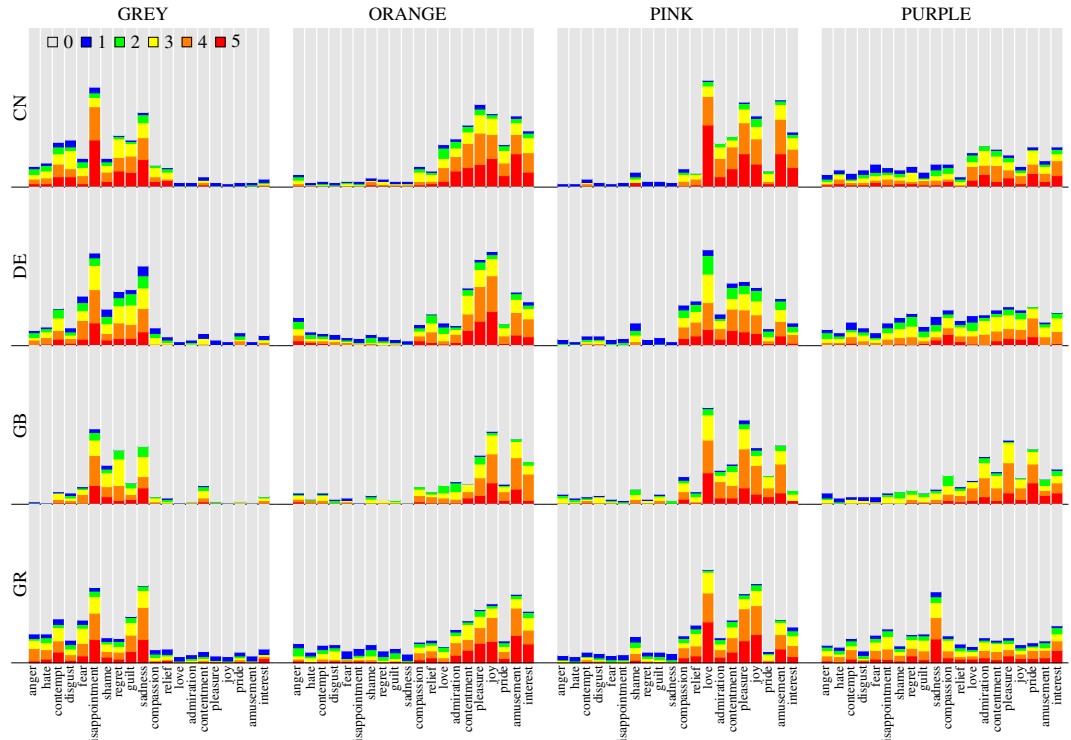

**Figure 5.** Colour terms GREY, ORANGE, PINK and PURPLE: stacked bar charts showing proportions of the six GEW rating categories, separately for the four countries. Same format as figure 4.

ROC curve (AUC) was 0.928 (80.2% correctly classified instances), which represents a moderately high accuracy of the classification. This accuracy is above the chance level of 25% correctly classified instances.

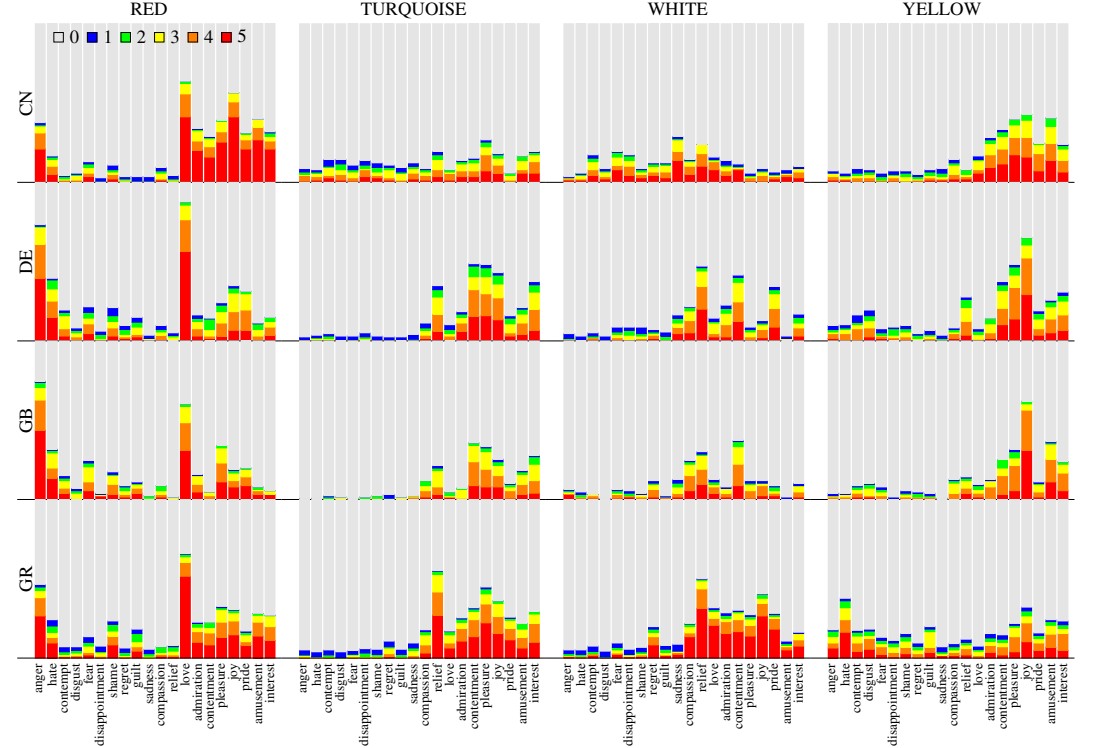

**Figure 6.** Colour terms RED, TURQUOISE, WHITE and YELLOW: stacked bar charts showing proportions of the six GEW rating categories, separately for the four countries. Same format as figure 4.

In comparison, the AUC of the classifier for datasets with randomly permuted class labels was approximately 0.5 (i.e. performance was at chance level). This latter value is significantly lower than the AUC of the classifier for the actual dataset ($p < 0.05$). These results indicate that our colour–emotion associations were relatively country-specific. Figure 7 shows that the proportion of participants for which the predicted country corresponded to the true country (i.e. the recall, shown on the main diagonal) was comparably higher for participants from China and Greece, and slightly lower for participants from Germany and the UK.

### 3.2.3. In-group advantage for the colour-classification

The second proposed measure for cultural differences in the colour–emotion associations is the *in-group advantage* when predicting the evaluated colour term from each set of 20 emotion association ratings. The same machine learning method as for the colour-classification across countries (above) was used. The main diagonals in the left part of table 3 show the colour-classification accuracy in terms of the proportion of correctly classified instances, separately for the four countries. For instance, for the data from Germany, the cross-validated proportion of correctly classified instances was 0.421.

The off-diagonal elements in the left part of table 3 show the classification accuracy when, for instance, a classifier trained on the data from Germany predicted the colour terms for the data from the UK. In this example, the proportion of correctly classified instances was 0.350. The right part of table 3 shows the differences between the within-country accuracy and the between-countries accuracy (i.e. the in-group advantage). For instance, the proportion of correctly classified instances was 0.087 higher when the data from Germany were predicted by a classifier trained on the data from Germany, compared with a classifier trained on the data from China. Table 3 shows that the in-group advantage for predicting the evaluated colour term from the 20 ratings of colour–emotion association ranged from 2% to 10% ($M = 6.1\%$, s.d. $= 2.2\%$). Thus, while the relatively high country-classification accuracy shows that colour–emotion associations are indeed country-specific to some degree, the relatively small in-group advantage for the colour classification demonstrates that colour–emotion associations are still relatively similar across countries.

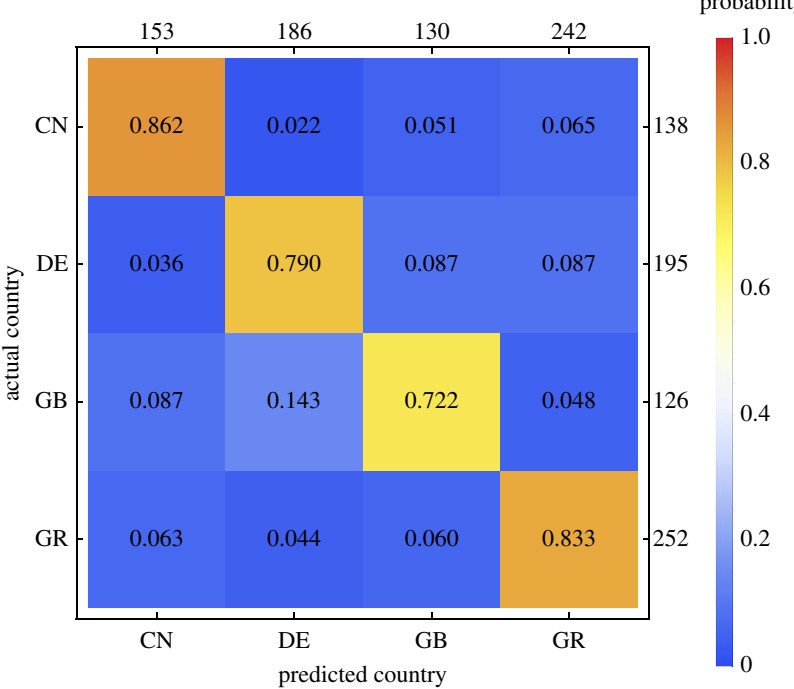

**Figure 7.** Confusion matrix for the prediction of the country of origin based on the 12 × 20 colour–emotion association ratings provided by each participant. Rows represent the actual and columns the predicted country of origin, respectively (CN: China, DE: Germany, GB: United Kingdom, GR: Greece). The number in each cell represents the probability that a participant originating from the country specified on the vertical axis (row) was classified as originating from the country specified on the horizontal axis (column). Thus, proportions on the main diagonal represent the true positive rate, or recall. The numbers on the right-hand side of the frame represent the absolute frequency of participants actually originating from a given country. The numbers on the upper side of the frame represent the absolute frequency of participants predicted to originate from a given country.

## 4. Discussion

In an online survey, 711 participants from China, Germany, Greece and the UK rated associations between 12 colour terms and 20 discrete emotions in their native languages. We used a machine learning approach to quantify the consistency and specificity of colour–emotion associations and their cultural specificity. The accuracy of a statistical classifier in decoding the colour evaluated on a given trial from the 20 ratings of colour–emotion association provides a measure of the consistency and specificity of colour–emotion associations. Across the 12 colour terms, the classifier accuracy was 38.7%. This value is above chance level (1/12, 8.3%), but not very high. Moreover, the accuracies ranged from 18% (TURQUOISE) to 68% (RED) (figure 3). These percentages indicate that colour–emotion associations are to some degree colour-specific, but do also speak against a one-to-one mapping between a colour term and an associated emotion.

The accuracy of a classifier in predicting the country of origin from the 240 individual colour–emotion associations provides a measure of the country-specificity of colour–emotion associations. The classification accuracy was significantly above chance level, demonstrating that the colour–emotion associations are to some degree country-specific. A second measure of the country-specificity is the in-group advantage of the colour-decoding accuracy. The proportion of correctly classified instances was at most 10% higher when the classifier was trained and tested on data from the same country, compared with when the classifier was trained on data from one country and then tested on data from a different country. Thus, we observed an in-group advantage, though relatively small, nevertheless indicating that colour–emotion associations are at the same time universal and culture-specific.

Taken together, our machine learning approach provided some interesting insights into the nature of colour–emotion associations. It demonstrated, for instance, that certain colour terms (e.g. BLACK and RED) were classified more accurately than others (e.g. GREEN, PURPLE and TURQUOISE). The difference in classification accuracy indicated that BLACK and RED evoked more consistent colour–

**Table 3.** Colour-classification accuracy within and across countries. The row label shows the country used for training the classifier. The column label shows the country on which the classifier was tested. Left part: proportion of correctly classified instances (recall). Entries on the main diagonal (italics) show the within-country proportion of correctly classified instances, based on 10-fold cross-validation. The other entries show the proportion of correctly classified instances when the classifier was trained on the data from the country specified by the row label and then tested on the data from the country specified by the column label. Right part: within-country proportion of correctly classified instances minus between-countries proportion of correctly classified instances. Positive values indicate an in-group advantage.

| | proportion of correctly classified instances | | | | accuracy (within)—accuracy (between) | | | |
| | test data | | | | test data | | | |
| training data | CN | DE | GR | GB | CN | DE | GR | GB |
| --- | --- | --- | --- | --- | --- | --- | --- | --- |
| CN | *0.337* | 0.334 | 0.270 | 0.321 | | 0.087 | 0.101 | 0.051 |
| DE | 0.290 | *0.421* | 0.311 | 0.350 | 0.047 | | 0.061 | 0.022 |
| GR | 0.274 | 0.362 | *0.371* | 0.309 | 0.063 | 0.059 | | 0.063 |
| GB | 0.279 | 0.384 | 0.289 | *0.372* | 0.058 | 0.037 | 0.082 | |

emotion association patterns than other colour terms. BLACK and RED colour terms were also more commonly associated with strong emotions, and evoked a larger number of associated emotions than many other colour terms. Here, and in the literature, one finds that RED is associated with both positive [10,21] and negative [21,44] emotions while BLACK is unambiguously associated with negative emotions [21,25,45]. In the current study, RED was often associated with love and anger, while BLACK was associated with sadness, hate and fear among other negative emotions. It is an interesting question whether the 'valence ambiguity' of RED contributes to inconsistent findings concerning cognitive effects of the colour red. Some authors proposed that an association between red and danger (or anger) can affect behaviour on cognitive tasks (e.g. [44,46,47]), while other studies failed to observe this effect (e.g. [48–51]).

The confusion matrix provided by the machine learning analysis can also be used to identify colour terms with similar emotional associations. For instance, the classifier was equally likely to classify ORANGE correctly as ORANGE or incorrectly as YELLOW (figure 3). This indicates that ORANGE and YELLOW colour terms carry shared emotional associations. YELLOW, however, was more often correctly classified as YELLOW than incorrectly classified as ORANGE. The latter observation indicates that certain emotional associations were unique to YELLOW and not shared with ORANGE. Thus, YELLOW seems to have a broader range of unique emotional associations than ORANGE, while ORANGE shares most of its emotional associations with YELLOW. A quantitative measure of the similarity between a pair of colours can be readily computed from the classifiers' confusion matrices, based on the assertion that colours that are more similar in terms of the associated emotions will be more often confused by the classifier than colours that are less similar. We used Luce's biased choice model (eq. 5 in [52]; see also [53]) to estimate similarity values for all pairs of colours (in terms of their emotion associations) from the confusion matrix shown in figure 3. By definition, the similarity between a colour and itself is 1.0 (representing maximum similarity), while 0.0 is the minimum possible similarity value, indicating that the emotion associations for the two colours are completely dissimilar. The similarity matrix is shown in figure 8. Particularly high similarity values were estimated for the pairs YELLOW–ORANGE, GREEN–TURQUOISE and BLUE–TURQUOISE.

Orange and yellow are perceptually similar colours, with similar hue angles in for instance the CIELAB colour space [54], and both are being considered as warm hues [32]. Perceptual similarity could potentially explain some of the overlap in emotional associations between ORANGE and YELLOW colour terms. Several other terms of perceptually similar colours were also frequently confused by the classifier (e.g. TURQUOISE was misclassified as GREEN or BLUE) but others were not (e.g. RED was not misclassified as ORANGE or PURPLE). This observation raises the question of how important perceptual similarity is for emotion associations with colour terms. In the online survey, we used linguistic colour terms to assess semantic associations as it is impossible to control the colorimetric coordinates of colours in an online survey. When presenting colour terms, we do not know if participants try to imagine these colours and which colorimetric coordinates these imagined

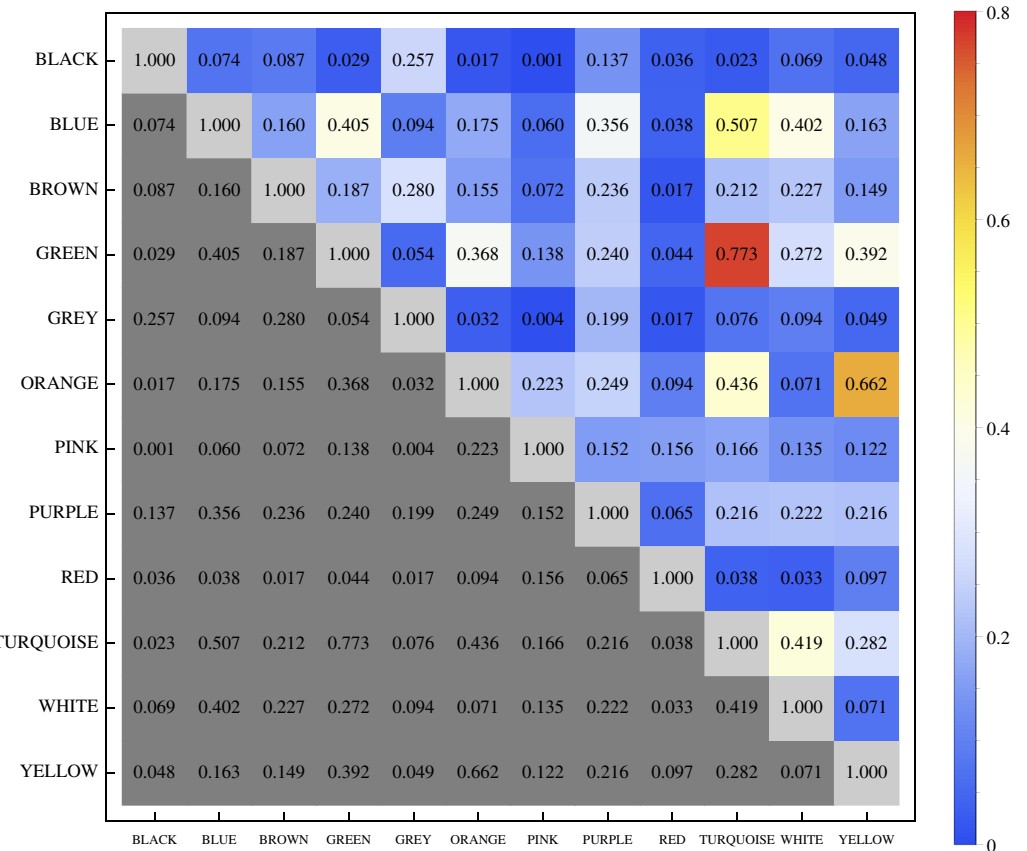

**Figure 8.** Matrix of similarity values for all pairs of colours, estimated with Luce's biased choice model [52] from the confusion matrix shown in figure 3. By definition, the similarity between one colour and itself is 1.0 and the minimum possible similarity is 0.0. Because the matrix is symmetric, only the upper triangle of the matrix needs to be considered and the remaining cells are shown in grey.

colours might have. One possibility is that participants imagine focal colours—the best exemplars of a given colour term [33]. Colorimetric coordinates of focal colours vary slightly between individuals [55] and across cultures [35] but they seem to converge on specific focal points [56]. Potentially, some participants might be picturing lighter colours (e.g. sky blue) and others darker colours (e.g. navy blue) for the same colour term (e.g. BLUE). This could especially be the case for Greek speakers who have an additional 12th basic colour term for LIGHT BLUE ('γαλάζιο') [57]. In the current study, the Greek term for LIGHT BLUE was assessed in the TURQUOISE category. Thus, the Greek participants might have pictured a darker shade of blue when given the colour term BLUE ('μπλε') than participants from the other countries. Hence, the imagined lightness can hardly be inferred from colour terms.

In general, colour terms do not encompass the entire gamut of colours. Colour terms mainly focus on the hue dimension (what laymen commonly refer to as colour—red, yellow, green, etc.) and pay little attention to lightness or brightness (perceived intensity of light; e.g. bright versus dim), and saturation or chroma (difference to an achromatic stimulus of the same brightness; e.g. neutral grey versus pure red). Thus, it is an empirical question if the observed associations between e.g. the colour term RED and the 20 discrete emotions are similar to the associations between a visually presented red stimulus and the same emotions. In addition to hue, lightness and chroma/saturation were reported to influence colour–emotion associations [3,26] and emotional responses to colour [58,59]. Our data show a high, but not perfect degree of similarity between the emotion associations for RED and PINK, and thus a potential effect of (imagined) lightness or saturation, because pink is a red hue with high lightness and low saturation. PINK seems to be exclusively associated with positive emotions while RED additionally carries negative associations with anger. It remains to be shown how colour-specific and culture-specific emotion associations are across a wide range of visually

presented colours varying in hue, saturation and lightness. The machine learning-based measures developed here can be applied to such data.

In the present study, explicit ratings of colour–emotion associations were collected. Colour-meaning associations (including colour–emotion associations) have also been found in studies using implicit measures of association, as for example in response time paradigms (e.g. [8,13,60,61]). However, we are not aware of studies measuring associations between colours and a wide range of discrete emotions with implicit tests. Thus, at present, it is not clear whether implicitly measured associations would show exactly the same patterns as the explicit ratings of colour–emotion associations collected in the present experiment. Our machine learning approach can also be used to analyse colour–emotion associations on the basis of implicit measures.

As described in the Methods section, the age and gender distributions were not identical between countries. The same observation applies to several previous studies interested in the link between colour and emotion, which analysed samples not balanced for gender (e.g. [21,25,29]), and often tested a relatively young (student) population (e.g. [21,25,29,59]). Here, we aimed to have a larger age range, and the age range was similar between the four countries (between 15 and 79 years old). Still, the differences in the age and gender distributions might have contributed to the differences in colour–emotion associations between countries, and larger and balanced datasets are needed to further investigate these issues. It should be noted that precisely the machine learning approach we propose here for quantifying differences between countries can also be used for quantifying differences in the colour–emotion associations between different age groups or different genders.

The machine learning approach provided further insight into cultural differences in colour–emotion associations. The classifier was able to use emotion associations with colour terms to classify countries above chance level, hinting at cultural specificity of certain colour–emotion associations (also see similar conclusions in [29], based on a smaller number of colour and emotion terms). For example, participants from China were more likely to associate negative emotions with WHITE than participants from the other three countries. In China, white is the colour commonly worn at funerals [62]. Thus, white might carry negative connotations due to its pairing with sad events in China, which is not the case in the European countries that were studied. Similarly, only Greek participants associated PURPLE with sadness while participants from other countries evaluated PURPLE as emotionally ambivalent (China and Germany) or positive (the UK). PURPLE is a liturgical colour symbolizing suffering, pain and mourning, and it is used for decorations during Advent and Lent in orthodox churches. The association between sadness and PURPLE might be more dominant in Greece than other studied countries, when considering that Greeks sometimes wear darker shades of purple during mourning periods. The origins of these cross-cultural differences in colour–emotion associations cannot be determined from the current study. Nevertheless, some cross-cultural differences could potentially be explained through the existence of different colour metaphors (conceptual metaphor theory, [13,63]), which would be semantically mediated, or different perceptual cross-modal associations between colours and emotional stimuli (statistical correspondence framework; [64]).

Furthermore, our approach demonstrated a detectable but small in-group advantage of on average 6.1% when classifying colours based on emotion associations. In-group advantages have been studied extensively in order to quantify the cultural specificity of the recognition of discrete emotions from facial expression (e.g. [31]). In this area, there is also evidence for both universality and cultural specificity of emotion recognition [65]. According to a meta-analysis [66], the accuracy (in terms of guessing-corrected per cent correct) in emotion recognition is on average 9.3% higher when emotions are both expressed and perceived by members of the same cultural group, compared with when the person who expresses the emotion and the person who perceives the emotion are from different cultural groups. The in-group advantage for colour decoding from emotion associations is thus somewhat lower than the in-group advantage found for emotion recognition from facial expression.

It should be noted that the reported classification accuracies represent a lower bound due to unavoidable imperfection of the statistical classifier. On a related note, the three measures we propose can be computed using any possible classification algorithm. Here, we used optimized support vector machines, which are known to perform relatively well but might be outperformed by more advanced methods, as for example deep artificial neural networks [67].

The machine learning method we propose is a novel approach in the field of colour and emotion, or in research on emotion more generally, but is well established in the neurosciences. For instance, in vision, the accuracy of statistical classifiers trained on fMRI data is used to evaluate whether, e.g., different pattern orientations cause specific activation patterns in the primary visual cortex [68]. Our methods

could also be helpful in other areas of research on emotion where multivariate datasets are available. For instance, if for a set of discrete emotions [69], several different physiological responses are measured (e.g. [70]), then our method can be used to quantify how specific the patterns of physiological responses are for the discrete emotions.

Taken together, the current study demonstrates how machine-learning techniques can be applied to study empirical questions about the psychology of colour and emotion. Using statistical classifiers, we could quantify the consistency and specificity of colour–emotion associations, as well as their cultural specificity. These analyses showed a moderate specificity and consistency of colour–emotion associations. Our data further indicated some degree of cultural specificity among our four countries—China, Germany, Greece and the UK. Colour–emotion associations were successfully classified when the algorithm was trained on a different country, but the classification was most accurate when the algorithm was trained on the country in question. We conclude that colour–emotion associations have both universal and culture specific features.

Ethics. The research was conducted according to the principles expressed in the Declaration of Helsinki. Participation in the online survey was voluntary, no potentially identifying information such as IP addresses was collected and participants were informed that (i) their answers would be treated anonymously, and (ii) they could unconditionally stop the participation. Informed consent was obtained from all subjects. In the Canton of Vaud, Switzerland, such questionnaire research does not require further ethical committee approval.

Data accessibility. The dataset supporting this article has been uploaded as part of the electronic supplementary material.

Authors' contributions. Conception and design of the experiment: N.D. and C.M. Conception of analysis method: D.O. Implementation of analysis methods: D.O. and J.W. Data analyses: D.O., J.W. and D.J. Development of the online survey, weblink hosting and support: N.D., D.J. and C.M. Data collection UK sample: J.H. Data collection Greek sample: M.P.-P. Data collection Chinese sample: M.Z. Data collection German sample: D.O. Manuscript writing (original draft): D.O., D.J. and C.M. Manuscript writing (editing): D.O., D.J., C.M., N.D., J.W., M.P.-P., M.Z. and J.H. All authors gave final approval for publication.

Competing interests. The authors declare that they have no competing interests.

Funding. The research was supported by the Institute of Psychology, University of Lausanne with a grant to C.M., AkzoNobel Imperial Chemical Industries (ICI) with a grant to C.M., and Swiss National Science Foundation (SNSF) with a Doc.CH fellowship grant to D.J. (P0LAP1_175055).

Acknowledgements. We thank Angeliki Theodoridou Lund, Afrodite Kapsaridi and Achilleas Papakonstantis for contributing to the translation of the survey instructions and the Geneva Emotion Wheel emotion terms. We are grateful to Amer Chamseddine for creating and to Yann Schrag for maintaining the online survey. We wish to thank AkzoNobel, Imperial Chemical Industries (ICI) Limited, in particular Dr David Elliott and Dr Tom Curwen, Color R&I team, Slough, UK, and Stephanie Kraneveld, Sassenheim, the Netherlands, for supporting our empirical work on colour preferences and emotions. Some of the figures for this article have been created using the SciDraw scientific figure preparation system (http://scidraw.nd.edu; [71]).

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
