## [Reviewer comments · Royal Society Open Science]

Review History

RSOS-181134.R0 (Original submission)

Review form: Reviewer 1 (Antonio Fernández-Caballero)

Is the manuscript scientifically sound in its present form?

Yes

Are the interpretations and conclusions justified by the results?

No

Is the language acceptable?

Yes

Is it clear how to access all supporting data?

Yes

Do you have any ethical concerns with this paper?

No

Have you any concerns about statistical analyses in this paper?

Yes

Recommendation?

Major revision is needed (please make suggestions in comments)

Comments to the Author(s)

This paper focuses on a transnational study of the specificity of color-emotion associations and their cultural differences. The problem faced is of great interest as results on the topic can be used in multiple domains (health, market, ...). Moreover, such approach involving four different countries (China, Germany, Greece and the UK) is always welcome!

The paper is well written (in general) and probably the efforts put in the project are worth while being published. Nonetheless, this reviewer feels that some work should be put in order to enhance the manuscript.

Firstly, some more discussion is needed to consolidate the overall results. Are 711 participants sufficient to validate the final results? The authors should discuss this issue considering that there are 240 potential color-emotion associations. Please consider not only the 240 associations, but also the fact that the questionnaire has been deployed in 4 countries.

Another key element that should be discussed is why these 4 specific countries have been chosen for a study involving cultural aspects. Are these four countries really representative of the most relevant cultures in the world? Please provide some comments on this!

The section dedicated to the participants offers Table 1 which shows some statistically unbalanced data: great differences between females and males, and mean age. This should not only be stated in the article but also discussed.

In page 16, the authors claim that the $AUC = 0.830$ (38.7% correctly classified instances) and that this corresponds to a moderately high classification accuracy. I do not agree. Please provide some more considerations!

Tables 4, 5 and 6 are described in terms of the values gotten, but the authors should also provide an explanation (if any) of the reasons behind the results obtained. Please do not only mention the similarities and differences among colors and countries, but also why you think this is the case.

Table 2 also shows that the data are quite bad! An explanation of the reason for these results have to be provided. Specifically, I would like to get some discussion of these results in relation to the "low" number of 711 participants (statistically speaking).

Lastly, I have some conflict with the following sentence from page 30: "Taken together, the current study demonstrates how machine learning techniques can be applied to study empirical questions about the psychology of color and emotion with complex data sets". In my opinion it is obvious that machine learning techniques are excellent in studying relations between emotions and color. But, here I would not say that we are in front of complex data sets.

In relation to the last paragraph of this review, I would recommend not to try to demonstrate the usefulness of the machine learning techniques implemented in this article (SVMs are more than accepted in the community) but on the psychological validity of the approach.

Review form: Reviewer 2

Is the manuscript scientifically sound in its present form?

Yes

Are the interpretations and conclusions justified by the results?

No

Is the language acceptable?

Yes

Is it clear how to access all supporting data?

Yes

Do you have any ethical concerns with this paper?

No

Have you any concerns about statistical analyses in this paper?

No

Recommendation?

Major revision is needed (please make suggestions in comments)

Comments to the Author(s)

This paper is a good attempt to investigate an interesting problem. The scope of the problem is very wide. Investigating the consistency of color - emotion relationship requires huge data and deep analysis. I think use of shallow machine learning approach is not sufficient. I am attaching file for more comments (Appendix A).

Decision letter (RSOS-181134.R0)

24-Oct-2018

Dear Dr Oberfeld,

The editors assigned to your paper ("A machine learning approach to quantifying the specificity of color-emotion associations and their cultural differences") have now received comments from reviewers. We would like you to revise your paper in accordance with the referee and Associate Editor suggestions which can be found below (not including confidential reports to the Editor). Please note this decision does not guarantee eventual acceptance.

Please submit a copy of your revised paper before 16-Nov-2018. Please note that the revision deadline will expire at 00.00am on this date. If we do not hear from you within this time then it will be assumed that the paper has been withdrawn. In exceptional circumstances, extensions

may be possible if agreed with the Editorial Office in advance. We do not allow multiple rounds of revision so we urge you to make every effort to fully address all of the comments at this stage. If deemed necessary by the Editors, your manuscript will be sent back to one or more of the original reviewers for assessment. If the original reviewers are not available, we may invite new reviewers.

- Data accessibility

<http://datadryad.org/submit?journalID=RSOS&manu=RSOS-181134>

- Competing interests

- Authors' contributions

- Acknowledgements

- Funding statement

Please note that Royal Society Open Science charge article processing charges for all new submissions that are accepted for publication. Charges will also apply to papers transferred to Royal Society Open Science from other Royal Society Publishing journals, as well as papers submitted as part of our collaboration with the Royal Society of Chemistry (<http://rsos.royalsocietypublishing.org/chemistry>). If your manuscript is newly submitted and subsequently accepted for publication, you will be asked to pay the article processing charge, unless you request a waiver and this is approved by Royal Society Publishing. You can find out more about the charges at <http://rsos.royalsocietypublishing.org/page/charges>. Should you have any queries, please contact openscience@royalsociety.org.

on behalf of Dr Narayanan Srinivasan (Associate Editor) and Prof. Antonia Hamilton (Subject Editor)
openscience@royalsociety.org

Associate Editor's comments (Dr Narayanan Srinivasan):

Two reviewers have commented on the paper. They suggest significant revisions. Authors are requested to address all the points in their revision.

Comments to Author:

Reviewers' Comments to Author:

Reviewer: 1

Comments to the Author(s)

This paper focuses on a transnational study of the specificity of color-emotion associations and their cultural differences. The problem faced is of great interest as results on the topic can be used

in multiple domains (health, market, ...). Moreover, such approach involving four different countries (China, Germany, Greece and the UK) is always welcome!

The paper is well written (in general) and probably the efforts put in the project are worth while being published. Nonetheless, this reviewer feels that some work should be put in order to enhance the manuscript.

Firstly, some more discussion is needed to consolidate the overall results. Are 711 participants sufficient to validate the final results? The authors should discuss this issue considering that there are 240 potential color-emotion associations. Please consider not only the 240 associations, but also the fact that the questionnaire has been deployed in 4 countries.

Another key element that should be discussed is why these 4 specific countries have been chosen for a study involving cultural aspects. Are these four countries really representative of the most relevant cultures in the world? Please provide some comments on this!

The section dedicated to the participants offers Table 1 which shows some statistically unbalanced data: great differences between females and males, and mean age. This should not only be stated in the article but also discussed.

In page 16, the authors claim that the AUC = 0.830 (38.7% correctly classified instances) and that this corresponds to a moderately high classification accuracy. I do not agree. Please provide some more considerations!

Tables 4, 5 and 6 are described in terms of the values gotten, but the authors should also provide an explanation (if any) of the reasons behind the results obtained. Please do not only mention the similarities and differences among colors and countries, but also why you think this is the case.

Table 2 also shows that the data are quite bad! An explanation of the reason for these results have to be provided. Specifically, I would like to get some discussion of these results in relation to the "low" number of 711 participants (statistically speaking).

Lastly, I have some conflict with the following sentence from page 30: "Taken together, the current study demonstrates how machine learning techniques can be applied to study empirical questions about the psychology of color and emotion with complex data sets". In my opinion it is obvious that machine learning techniques are excellent in studying relations between emotions and color. But, here I would not say that we are in front of complex data sets.

In relation to the last paragraph of this review, I would recommend not to try to demonstrate the usefulness of the machine learning techniques implemented in this article (SVMs are more than accepted in the community) but on the psychological validity of the approach.

Reviewer: 2

Comments to the Author(s)

This paper is a good attempt to investigate an interesting problem. The scope of the problem is very wide. Investigating the consistency of color - emotion relationship requires huge data and deep analysis. I think use of shallow machine learning approach is not sufficient. I am attaching file for more comments.

Author's Response to Decision Letter for (RSOS-181134.R0)

See Appendix B.

RSOS-181134.R1 (Revision)

Review form: Reviewer 1 (Antonio Fernández-Caballero)

Is the manuscript scientifically sound in its present form?

Yes

Are the interpretations and conclusions justified by the results?

No

Is the language acceptable?

Yes

Is it clear how to access all supporting data?

Yes

Do you have any ethical concerns with this paper?

No

Have you any concerns about statistical analyses in this paper?

Yes

Recommendation?

Reject

Comments to the Author(s)

After my first review my decision was Major Review.

Unfortunately, the authors have not responded satisfactorily to some of the most important drawbacks. Rather they have tried to demonstrate that my comments were not pertinent.

Decision letter (RSOS-181134.R1)

20-Feb-2019

Dear Dr Oberfeld:

I write you in regards to manuscript # RSOS-181134.R1 entitled "A machine learning approach to quantifying the specificity of color-emotion associations and their cultural differences" which you submitted to Royal Society Open Science.

Regrettably, in view of the criticisms of the reviewer(s) found at the bottom of this letter, your manuscript has been denied publication in Royal Society Open Science.

Thank you for considering Royal Society Open Science for the publication of your research. I hope the outcome of this specific submission will not discourage you from the submission of future manuscripts.

on behalf of Dr Narayanan Srinivasan (Associate Editor) and Antonia Hamilton (Subject Editor)
openscience@royalsociety.org

Subject Editor comments:

I have read the original reviews in detail. Both reviewers commented that the dataset was small and non-exhaustive, and your reply was that the machine learning component of the paper was only exploratory. But without testing the machine learning algorithms on an independent sample, it is not clear what you can conclude - much of the good performance might just be overfitting. For this reason, I agree with the reviewer and associate editor comments below that this paper should be rejected.

Associate Editor Comments to Author (Dr Narayanan Srinivasan):

We apologise for the unusual time taken to complete peer review: unfortunately, the journal was only able to secure the further support of one of the original reviewers, despite repeated attempts to contact the second reviewer. To avoid further delays, we have decided to make a decision on the one reviewer who has reported on this revision. Regrettably, they do not consider that the changes you have made change their original view, and that additional work would be required before publication - in-line with the journal's policy of not generally permitting more than one round of major revision, we must therefore reject your paper on this occasion. We hope the feedback from the referees proves useful if you choose to submit the research elsewhere -- good luck!

Reviewer comments to Author:

Reviewer: 1

Comments to the Author(s)

After my first review my decision was Major Review.

Unfortunately, the authors have not responded satisfactorily to some of the most important drawbacks. Rather they have tried to demonstrate that my comments were not pertinent.

Author's Response to Decision Letter for (RSOS-181134.R1)

See Appendix C.

RSOS-190741.R0

Review form: Reviewer 3

Is the manuscript scientifically sound in its present form?

Yes

Are the interpretations and conclusions justified by the results?

Yes

Is the language acceptable?

Yes

Is it clear how to access all supporting data?

Yes

Do you have any ethical concerns with this paper?

No

Have you any concerns about statistical analyses in this paper?

No

Recommendation?

Accept with minor revision (please list in comments)

Comments to the Author(s)

Apologies, I missed that this was a resubmission with my initial review. I have looked at their response (I don't have access to the original reviews) and will discuss the three points that they discuss here.

- Complaints about sample size- I agree with the authors that their sample size of 711 participants is more than adequate in this context. I also agree that in comparison to a large 'big data' set, 711 is 'small'. However, the current sample size is bigger than is often found in many studies of colour-emotion (or more generally in studies of colour cognition), and so makes an important contribution to the field. Having looked at the task (via the link which the authors make available in text) I think it is appropriate for the research questions being asked, and their screening of participant data allows for confidence in the responses that they acquired.

- Classifier accuracy - The unique contribution of this study is that it uses an established method in a novel way. In many parts of colour cognition, researchers are rarely looking for a single variable which can account for all the variance in their data.

Consider for example the field of colour preference - there are many different variables which have been shown to influence colour preference, just some of which are listed below:-

ease of naming of colours and prototypicality of colour; ecological valence theory (that objects are liked as a result of the interaction between either experience over the lifetime with objects of that colour, or potentially over the course of evolution); colour preference in children has been tied to developing an identity of the self (pink for girls, blue for boys); relationship with the S/(L+M) and L/(L+M) pathways of colour vision.

As a result, most models of colour preference account for a proportion of variance within data (and vice versa, would *predict* a proportion of the variance) - no single model would expect to account for all the variance in a dataset, although clearly some will account for more than others (say, 70% vs 30%). This *amount* that can be explained is informative in understanding the hierarchy of these variables in a colour perception task. The same is true in studies of colour-emotion. The valuable insight gained from the data analysis in the current paper is that there is a metric which will allow for quantification of the strength of different variables impact on colour-emotion associations.

Robustness of the model : I agree with the points the authors raise in their response - if the concern here is related to the previous two points (sample size and accuracy), then I think concerns about inaccuracy aren't correct. There is always, as with all cognitive based models, some room for error - the authors make this clear in their manuscript. The extra data mentioned, from more countries, adds to the authors point here.

** Initial review submitted 01/05/19 - these suggested changes are very minor, and I think the paper could be accepted without significant, or possibly any, changes to these points. Apologies to the authors- had I realised the initial comments I would have focused my review on those points.

This was a really interesting paper providing interesting insights using a novel method into colour-emotion associations. The data itself is very rich, providing both qualitative and quantitative insights into the relationship of colour-emotion, and there is a lot of ground covered in terms of scope and literature.

My comments are very minor. I expect that that these changes would be at most a handful of sentences, but would strengthen some of the arguments put forward in the discussion in particular -

1. Discussion - line 498 there is some discussion about perceptual similarity. It's a bit unclear what is really meant by perceptual similarity here. I'd be cautious, for example, about using the phrase "orange being a mixture of red and yellow" - there are models of colour which are not additive in this way after all, where orange could be described in terms of e.g. cone excitation, and so independent of the red and yellow hues, but still perceptually similar. This can be rectified by expanding very slightly (by maybe one sentence) on what perceptual similarity means here, or just removing the phrase above.

2. Discussion - around 507. There's a great discussion of the relationship between focal colour terms and the relationships reported in the paper. I agree with the authors about the convergence of focal terms in the colour space, but do have a remaining question about the colour terms they've used. For example, Greek - one of the languages chosen, I believe has two basic colour terms for blue: two BCTs for blue, $\mu \pi \lambda \epsilon$ [blé] "blue" and $\gamma \alpha \lambda \acute{\alpha} \zeta \iota \omicron$ [yalazjo] "light blue", meaning all of English BCTs are covered in the study, but not all Greek BCTs. Any differences between BCTs in these languages should be acknowledged at this point in the discussion. It doesn't threaten any of their findings - the authors are clear on the variability in e.g. focal terms across participants and languages, but it may be helpful to flag this to readers. One or two sentences at most.

3. Discussion- around 559. Although the authors do discuss cultural differences in use of colour - it still doesn't tell us about how exactly these associations are learned and the mechanisms behind it. e.g. is it semantic mediation? I appreciate the results of this study can't tell us this, but a single sentence or reference to this would help the reader put this research into context.

Review form: Reviewer 4

Is the manuscript scientifically sound in its present form?

Yes

Are the interpretations and conclusions justified by the results?

Yes

Is the language acceptable?

Yes

Is it clear how to access all supporting data?

No

Do you have any ethical concerns with this paper?

No

Have you any concerns about statistical analyses in this paper?

No

Recommendation?

Major revision is needed (please make suggestions in comments)

Comments to the Author(s)

When I accepted the review, I was not aware that this manuscript had been previously reviewed. I was therefore surprised to see a rebuttal letter from the authors. This letter addressed most of my more serious potential concerns very well. I will therefore only focus on additional issues.

The topic is interesting and warrants more research. This paper seems a good first step in that direction.

The introduction is engagingly written, but somewhat lacking in rationale. For example, the authors propose an ingroup advantage in color-decoding accuracy but (a) they do not test color decoding accuracy, but rather color-emotion associations, (b) the proposed reference (31) has little to say about ingroup advantages and more importantly does not refer to color at all. If the authors want to use this concept in some sort of metaphorical manner, they need to explain why they think this to be helpful.

I understand that given the authors' goals sample size can be considered adequate. However, the samples are heavily based towards women. Given that stereotypical color preferences differ between the sexes I think this poses a problem in terms of interpretation.

Why were the color terms translated? This is an online study - why not show the colors?

Participants had to make 240 fairly demanding decisions in an online study. The authors do not report any attempt to check for attention. I would expect participants to fatigue and start giving more random answers. Also, in online studies it is never clear what else participants are doing at the same time. Attention checks are typically used and generally suggested for this type of study, especially when the task is long and tedious.

The discussion is very long and verbose and much of it repeats the results section. There is even a results figure in the discussion. What is missing is a more solid theoretical integration.

Also, while I understand that the authors consider their approach a first step with the goal to show that there is a consistent signal in color-emotion associations, I was missing an idea where to go from here.

Decision letter (RSOS-190741.R0)

14-Jun-2019

Dear Dr Oberfeld,

The Subject Editor assigned to your paper ("A machine learning approach to quantifying the specificity of color-emotion associations and their cultural differences") has now received comments from reviewers. We would like you to revise your paper in accordance with the referee and Associate Editor suggestions which can be found below (not including confidential reports to the Editor). Please note this decision does not guarantee eventual acceptance.

Please submit a copy of your revised paper before 07-Jul-2019. Please note that the revision deadline will expire at 00.00am on this date. If we do not hear from you within this time then it will be assumed that the paper has been withdrawn. In exceptional circumstances, extensions may be possible if agreed with the Editorial Office in advance. We do not allow multiple rounds of revision so we urge you to make every effort to fully address all of the comments at this stage. If deemed necessary by the Editors, your manuscript will be sent back to one or more of the original reviewers for assessment. If the original reviewers are not available we may invite new reviewers.

When submitting your revised manuscript, you must respond to the comments made by the referees and upload a file "Response to Referees" in "Section 6 - File Upload". Please use this to document how you have responded to each of the comments, and the adjustments you have made. In order to expedite the processing of the revised manuscript, please be as specific as possible in your response.

- Ethics statement

- Data accessibility

It is a condition of publication that all supporting data are made available either as supplementary information or preferably in a suitable permanent repository. The data

accessibility section should state where the article's supporting data can be accessed. This section should also include details, where possible of where to access other relevant research materials such as statistical tools, protocols, software etc can be accessed. If the data has been deposited in an external repository this section should list the database, accession number and link to the DOI for all data from the article that has been made publicly available. Data sets that have been deposited in an external repository and have a DOI should also be appropriately cited in the manuscript and included in the reference list.

If you wish to submit your supporting data or code to Dryad (<http://datadryad.org/>), or modify your current submission to dryad, please use the following link:
<http://datadryad.org/submit?journalID=RSOS&manu=RSOS-190741>

- **Competing interests**

- **Authors' contributions**

- **Acknowledgements**

- **Funding statement**

on behalf of Dr Narayanan Srinivasan (Associate Editor) and Antonia Hamilton (Subject Editor)
openscience@royalsociety.org

Associate Editor Comments to Author (Dr Narayanan Srinivasan):

Two reviewers have commented on the paper. While both find the work interesting, they have questions and suggestions to improve the paper. The authors are kindly requested to address all the comments and submit the revision.

Reviewer comments to Author:

Reviewer: 3

Comments to the Author(s)

Apologies, I missed that this was a resubmission with my initial review. I have looked at their response (I don't have access to the original reviews) and will discuss the three points that they discuss here.

- Complaints about sample size- I agree with the authors that their sample size of 711 participants is more than adequate in this context. I also agree that in comparison to a large 'big data' set, 711 is 'small'. However, the current sample size is bigger than is often found in many studies of colour-emotion (or more generally in studies of colour cognition), and so makes an important contribution to the field. Having looked at the task (via the link which the authors make available in text) I think it is appropriate for the research questions being asked, and their screening of participant data allows for confidence in the responses that they acquired.

- Classifier accuracy - The unique contribution of this study is that it uses an established method in a novel way. In many parts of colour cognition, researchers are rarely looking for a single variable which can account for all the variance in their data.

Consider for example the field of colour preference - there are many different variables which have been shown to influence colour preference, just some of which are listed below:- ease of naming of colours and prototypicality of colour; ecological valence theory (that objects are liked as a result of the interaction between either experience over the lifetime with objects of that colour, or potentially over the course of evolution); colour preference in children has been tied to developing an identity of the self (pink for girls, blue for boys); relationship with the S/(L+M) and L/(L+M) pathways of colour vision.

As a result, most models of colour preference account for a proportion of variance within data (and vice versa, would *predict* a proportion of the variance) - no single model would expect to account for all the variance in a dataset, although clearly some will account for more than others (say, 70% vs 30%). This *amount* that can be explained is informative in understanding the hierarchy of these variables in a colour perception task. The same is true in studies of colour-emotion. The valuable insight gained from the data analysis in the current paper is that there is a metric which will allow for quantification of the strength of different variables impact on colour-emotion associations.

Robustness of the model : I agree with the points the authors raise in their response - if the concern here is related to the previous two points (sample size and accuracy), then I think concerns about inaccuracy aren't correct. There is always, as with all cognitive based models, some room for error - the authors make this clear in their manuscript. The extra data mentioned, from more countries, adds to the authors point here.

** Initial review submitted 01/05/19 - these suggested changes are very minor, and I think the paper could be accepted without significant, or possibly any, changes to these points. Apologies

to the authors- had I realised the initial comments I would have focused my review on those points.

This was a really interesting paper providing interesting insights using a novel method into colour-emotion associations. The data itself is very rich, providing both qualitative and quantitative insights into the relationship of colour-emotion, and there is a lot of ground covered in terms of scope and literature.

My comments are very minor. I expect that that these changes would be at most a handful of sentences, but would strengthen some of the arguments put forward in the discussion in particular -

1. Discussion - line 498 there is some discussion about perceptual similarity. It's a bit unclear what is really meant by perceptual similarity here. I'd be cautious, for example, about using the phrase "orange being a mixture of red and yellow" - there are models of colour which are not additive in this way after all, where orange could be described in terms of e.g. cone excitation, and so independent of the red and yellow hues, but still perceptually similar. This can be rectified by expanding very slightly (by maybe one sentence) on what perceptual similarity means here, or just removing the phrase above.

2. Discussion - around 507. There's a great discussion of the relationship between focal colour terms and the relationships reported in the paper. I agree with the authors about the convergence of focal terms in the colour space, but do have a remaining question about the colour terms they've used. For example, Greek - one of the languages chosen, I believe has two basic colour terms for blue: two BCTs for blue, $\mu \pi \lambda \epsilon$ [blé] "blue" and $\gamma \alpha \lambda \acute{\alpha} \zeta \iota \omicron$ [yalazjo] "light blue", meaning all of English BCTs are covered in the study, but not all Greek BCTs. Any differences between BCTs in these languages should be acknowledged at this point in the discussion. It doesn't threaten any of their findings - the authors are clear on the variability in e.g. focal terms across participants and languages, but it may be helpful to flag this to readers. One or two sentences at most.

3. Discussion- around 559. Although the authors do discuss cultural differences in use of colour - it still doesn't tell us about how exactly these associations are learned and the mechanisms behind it. e.g. is it semantic mediation? I appreciate the results of this study can't tell us this, but a single sentence or reference to this would help the reader put this research into context.

Reviewer: 4

Comments to the Author(s)

When I accepted the review, I was not aware that this manuscript had been previously reviewed. I was therefore surprised to see a rebuttal letter from the authors. This letter addressed most of my more serious potential concerns very well. I will therefore only focus on additional issues.

The topic is interesting and warrants more research. This paper seems a good first step in that direction.

The introduction is engagingly written, but somewhat lacking in rationale. For example, the authors propose an ingroup advantage in color-decoding accuracy but (a) they do not test color decoding accuracy, but rather color-emotion associations, (b) the proposed reference (31) has little to say about ingroup advantages and more importantly does not refer to color at all. If the authors want to use this concept in some sort of metaphorical manner, they need to explain why they think this to be helpful.

I understand that given the authors' goals sample size can be considered adequate. However, the samples are heavily based towards women. Given that stereotypical color preferences differ between the sexes I think this poses a problem in terms of interpretation.

Why were the color terms translated? This is an online study – why not show the colors?

Participants had to make 240 fairly demanding decisions in an online study. The authors do not report any attempt to check for attention. I would expect participants to fatigue and start giving more random answers. Also, in online studies it is never clear what else participants are doing at the same time. Attention checks are typically used and generally suggested for this type of study, especially when the task is long and tedious.

The discussion is very long and verbose and much of it repeats the results section. There is even a results figure in the discussion. What is missing is a more solid theoretical integration.

Also, while I understand that the authors consider their approach a first step with the goal to show that there is a consistent signal in color-emotion associations, I was missing an idea where to go from here.

Author's Response to Decision Letter for (RSOS-190741.R0)

See Appendix D.

RSOS-190741.R1 (Revision)

Review form: Reviewer 3

Is the manuscript scientifically sound in its present form?

Yes

Are the interpretations and conclusions justified by the results?

Yes

Is the language acceptable?

Yes

Do you have any ethical concerns with this paper?

No

Recommendation?

Accept as is

Comments to the Author(s)

Thanks for your responses to my comments. I'm happy with the changes you've made to this paper, and think it should now be accepted.

Review form: Reviewer 4

Is the manuscript scientifically sound in its present form?

Yes

Are the interpretations and conclusions justified by the results?

Yes

Is the language acceptable?

Yes

Do you have any ethical concerns with this paper?

No

Recommendation?

Accept as is

Comments to the Author(s)

The response to my concerns can be considered adequate

Decision letter (RSOS-190741.R1)

22-Aug-2019

Dear Dr Oberfeld,

I am pleased to inform you that your manuscript entitled "A machine learning approach to quantifying the specificity of color-emotion associations and their cultural differences" is now accepted for publication in Royal Society Open Science.

on behalf of Dr Narayanan Srinivasan (Associate Editor) and Antonia Hamilton (Subject Editor)
openscience@royalsociety.org

Associate Editor Comments to Author (Dr Narayanan Srinivasan):

Associate Editor: 1

Comments to the Author:

Both reviewers are happy with the revisions you have made and are satisfied. We are happy to inform that the paper is now accepted for publication in RSOS.

Reviewer comments to Author:

Reviewer: 3

Comments to the Author(s)

Thanks for your responses to my comments. I'm happy with the changes you've made to this paper, and think it should now be accepted.

Reviewer: 4

Comments to the Author(s)

The response to my concerns can be considered adequate

Appendix A

The work is very interesting and organised well. The set objective of the work is to propose a machine learning approach to a) the consistency and specificity of color - emotion associations and b) the degree to which they are country-specific.

In my view the machine learning approach adopted here doesn't establish any of the two set objectives. Let me enumerate some of the reasons (although the author has not given any reasoning and the reasons are debatable) :

1. The machine learning approach works statistically on the quantifiable limited dataset provided during the learning. In this case there are 12 color terms and 20 discrete emotions. Both the sets are small and non-exhaustive. Various ML architectures and approaches may establish various types of relations. One can say that there exist a relation, but it requires separate validation to claim for a specific relationship and one should explain the reasons for existence of such a relationship. Testing phase validation and accuracy numbers in ML can not be treated as validation of the real world knowledge.
2. The Machine learning (ML) approach normally works with closed world assumption, i.e. only stated true statements (training set) are true. Hence, there is no difference between false and not stated statements. However, deep learning methods are taking up large data sets and claiming enough generalisation, but in this work no such claim exists. Author themselves have discussed many confusions existing, say for color RED. The prediction going beyond some chance level, can not be claimed as accurate.
3. In my opinion the color specificity of emotion and again country-specificity is culturally determined and requires analysis of knowledge available in languages. It also requires for taking context into consideration, say for example, RED for a soldier at war field, RED for a person driving vehicle, or RED in the flag of a country, can not be related to the same emotion.

Hence, this paper requires more debate and raises many unanswered questions.

Dr Narayanan Srinivasan
Royal Society Open Science

Department of Psychology
Section Experimental Psychology

Dr. Daniel Oberfeld, Ph.D.
Associate Professor

Johannes Gutenberg-Universität Mainz

Wallstrasse 3
Room 06-416
D-55122 Mainz
Germany

Tel. +49 6131 39-39274
Fax +49 6131 39-39268

23. November 2018

Revised manuscript "A machine learning approach to quantifying the specificity of color-emotion associations and their cultural differences" (RSOS-181134)

oberfeld@uni-mainz.de

<http://www.staff.uni-mainz.de/oberfeld/>

Dear Editor,

we are grateful to you and the reviewers for the positive evaluation of our manuscript and for the helpful comments.

We are now submitting a revised version for which we carefully considered all comments and implemented the suggested changes. We marked the edits in blue ink in the revised manuscript.

Our answers to the comments in the decision letter are listed below. We hope that the changes have made the paper acceptable for publication in your journal.

Best regards

Daniel Oberfeld (on behalf of all authors)

Response (indicated by bullets and blue ink) to the comments of Reviewer 1

This paper focuses on a transnational study of the specificity of color-emotion associations and their cultural differences. The problem faced is of great interest as results on the topic can be used in multiple domains (health, market, ...). Moreover, such approach involving four different countries (China, Germany, Greece and the UK) is always welcome!

The paper is well written (in general) and probably the efforts put in the project are worth while being published.

- Thank you very much for the positive evaluation of our paper.

Nonetheless, this reviewer feels that some work should be put in order to enhance the manuscript.

Firstly, some more discussion is needed to consolidate the overall results. Are 711 participants sufficient to validate the final results? The authors should discuss this issue considering that there are 240 potential color-emotion associations. Please consider not only the 240 associations, but also the fact that the questionnaire has been deployed in 4 countries.

- While the number of instances is arguably relatively small compared to "big data" applications of machine learning, it is important to note that we do not use the models to predict classes but rather to carry out an explorative analysis. We evaluated the significance of the results using randomization tests that proved that the results are statistically sound. Additionally, the data set is sparse, only few emotions are associated with a color, most emotion ratings are 0 (see our Figure 2). Finally, we want to point out that these dimensions are typical for applications of machine learning in science. Many medical applications still cope with much smaller number of instances, similar to chemistry, or the neurosciences (e.g., Haynes & Rees, 2005).

Another key element that should be discussed is why these 4 specific countries have been chosen for a study involving cultural aspects. Are these four countries really representative of the most relevant cultures in the world? Please provide some comments on this!

- We appreciate this comment. Indeed, one could wonder whether we had aimed to compare these four countries because we consider them to represent major cultures. No, we do not assume that these are representative for the variety of cultures we can see worldwide. We included Germany because the senior (last) author is from Germany and has collected the data there. We also wanted to include one culture (China in the current case) that differs from the frequently tested Western cultures (e.g. UK in the present case, US). Moreover, we were able to add a sample from Greece, which provided us with the opportunity of testing another sample that also belongs to the European continent, but where the Latin alphabet is not used (German and English words can be very similar in many instances). We felt that this data set is very useful to evaluate the usefulness of the ML approach to quantify similarities and differences in color-emotion associations between countries. Needless to say, this is the beginning of a larger research programme rather than a final outcome. It would indeed be very interesting to collect similar data in additional countries, and the machine learning methods proposed here will be helpful to analyze these data.
- In the revised manuscript we are now reporting: *"We selected these four countries because we had convenient access to study participants, and to include two European countries that were studied relatively often in the color-emotion literature (the UK and Germany; e.g., [27]), one Eastern culture (e.g., [29]), and one European country that has been studied less often (Greece) and also uses a non-Latin alphabet. We do not imply that these four countries are representative for all countries in the world. It would be interesting to collect data in additional countries, and the machine-learning approach to studying color-emotion associations we propose here can of course be applied to data sets including a higher number of countries."*

The section dedicated to the participants offers Table 1 which shows some statistically unbalanced data: great differences between females and males, and mean age. This should not only be stated in the article but also discussed.

- We now added a discussion of potential consequences of the differences in the age and gender distribution across countries: *"As described in the Method section, the age and gender distributions were not identical between countries. The same observation applies to several previous studies on color-emotion, which analyzed samples not balanced for gender (e.g., [20, 24, 27]), and often tested a relatively young (student) population (e.g., [20, 24, 27, 53]). Here, we aimed to have a larger age range, and the age range was similar between the four countries (between 16 and 79 years old). Still, the differences in the age and gender distributions might have contributed to the differences in color-emotion associations between countries, and larger and balanced datasets are needed to further investigate these issues. It should be noted that precisely the machine learning approach we propose here for quantifying differences between countries can also be used for quantifying differences in the color-emotion associations between different age groups or different genders."*

In page 16, the authors claim that the AUC = 0.830 (38.7% correctly classified instances) and that this corresponds to a moderately high classification accuracy. I do not agree. Please provide some more considerations!

- We agree that for classification purposes, this is not a high AUC, but we do not aim to carry out predictive machine learning, but rather an explorative analysis. In this context, as discussed in the paper, an AUC of 0.83 tells us that there is a signal in the data and we can pick it up and use the findings as indications for color-emotion relations in the data. Also we

want to point out that this is a multi-class problem with 12 classes (colors), therefore an accuracy of 38.7 is well above base performance ($1/12 = 0.0833$), backed up by the results of the randomization tests. As described in our paper, the fact that the accuracy of the classifier is significantly above chance performance shows that a systematic relation between color and associated emotions exists. At the same time, the relatively low classifier accuracy tells us that this is far from being a one-to-one relationship. Put differently, the limited accuracy does not imply that our data are "bad", as you put it in one of your following comments. Instead, it reveals an important and meaningful feature of the color-emotion associations, as we discuss in the paper.

Tables 4, 5 and 6 are described in terms of the values gotten, but the authors should also provide an explanation (if any) of the reasons behind the results obtained. Please do not only mention the similarities and differences among colors and countries, but also why you think this is the case.

- We had decided to restrict our discussion of the potential reasons for the cross-cultural similarities and differences to a minimum in this paper and to focus on the ML-based quantitative measures. However, following your suggestion, we now added a discussion of some of the most prominent cultural differences to the paper: *"For example, participants from China were more likely to associate negative emotions with WHITE than participants from the other three countries. In China, white is the color commonly worn at funerals [56]. Thus, white might carry negative connotations due to its pairing with sad events in China, which is not the case in the European countries that were studied. Similarly, only Greek participants associated PURPLE with sadness while participants from other countries evaluated PURPLE as emotionally ambivalent (China and Germany) or positive (the UK). PURPLE is a liturgical color symbolizing suffering, pain and mourning, and it is used for decorations during Advent and Lent in orthodox churches. The association between sadness and purple might be more dominant in Greece than other studied countries, also considering that Greeks sometimes wear darker shades of purple during mourning periods."*

Table 2 also shows that the data are quite bad! An explanation of the reason for these results have to be provided. Specifically, I would like to get some discussion of these results in relation to the "low" number of 711 participants (statistically speaking).

- As stated above, the limited accuracy does not imply that our data are "bad" but shows that human color-emotion associations are systematic but far from following a one-to-one relationship. As we already discuss in the paper, some portion of the relatively low classification accuracy can be attributed to (unavoidable) imperfections of the classifier. If a much larger dataset was available, more powerful methods like deep learning might produce somewhat higher accuracies. We believe, however, that the largest share of the limitation in classifier accuracy is due to *psychological* reasons (i.e., the same emotions are associated with multiple colors, not all participants report the same color-emotion associations) rather than due to imperfection of the statistical classifier.

Lastly, I have some conflict with the following sentence from page 30: "Taken together, the current study demonstrates how machine learning techniques can be applied to study empirical questions about the psychology of color and emotion with complex data sets". In my opinion it is obvious that machine learning techniques are excellent in studying relations between emotions and color. But, here I would not say that we are in front of complex data sets.

- We deleted this passage.

In relation to the last paragraph of this review, I would recommend not to try to demonstrate the usefulness of the machine learning techniques implemented in this article (SVMs are more than accepted in the community) but on the psychological validity of the approach.

- We of course agree that SVMs are very accepted in machine learning in general, but to our knowledge, they have never been applied to study color-meaning associations.

Reviewer: 2

This paper is a good attempt to investigate an interesting problem. The scope of the problem is very wide. Investigating the consistency of color - emotion relationship requires huge data and deep analysis. I think use of shallow machine learning approach is not sufficient.

The work is very interesting and organised well. The set objective of the work is to propose a machine learning approach to a) the consistency and specificity of color - emotion associations and b) the degree to which they are country-specific. In my view the machine learning approach adopted here doesn't establish any of the two set objectives.

Let me enumerate some of the reasons (although the author has not given any reasoning and the reasons are debatable) :

1. The machine learning approach works statistically on the quantifiable limited dataset provided during the learning. In this case there are 12 color terms and 20 discrete emotions. Both the sets are small and non-exhaustive. Various ML architectures and approaches may establish various types of relations. One can say that there exist a relation, but it requires separate validation to claim for a specific relationship and one should explain the reasons for existence of such a relationship. Testing phase validation and accuracy numbers in ML can not be treated as validation of the real world knowledge.

- As discussed above in response to a comment by Reviewer 1, in this research we do not use machine learning to provide models that perform well when predicting, but rather as an explorative tool, where the predictions of the classifiers are used to find indications for psychological relationships between color and emotion. As pointed out above, the fact that the accuracy of the classifier is significantly above chance performance shows that a systematic relation between color and associated emotions exists. At the same time, the relatively low classifier accuracy tells us that this is far from being a one-to-one relationship. This is discussed in our paper in relation to other research studies. Additionally, looking at the statistics of the data set, we can see that it is sparse, which makes sense as colors are not associated with too many emotions, but rather a small subset. This means the values are well distributed over all attributes, limiting any problem that could arise from having too many attributes and too few instances. In addition, while big data relies on larger and more complex data sets, machine-learning can very well cope with smaller and less complex data, as many examples from medicine or neuroscience show. Using randomization tests, we also showed that the classifier accuracy is significantly higher than chance performance.

2. The Machine learning (ML) approach normally works with closed world assumption, i.e. only stated true statements (training set) are true. Hence, there is no difference between false and not stated statements. However, deep learning methods are taking up large data sets and claiming enough generalisation, but in this work no such claim exists. Author themselves have discussed many confusions existing, say for color RED. The prediction going beyond some chance level, can not be claimed as accurate.

- First, as discussed above, we do not claim perfect accuracy but discuss that there are both distinct and similar emotion associations for the colors, and that some pairs of colors show higher similarity than other pairs. We of course agree that the list of 20 discrete emotions was not exhaustive. Still, it represents an established selection of emotions viewed as being among the most important according to the emotion literature. It is also a larger set of emotions than the sets included in several previous studies on cross-cultural color-emotion associations. Second, while it is true that deep learning uses (and requires) larger data sets, this is not true for machine learning in general, which is an area that covers much more than just deep learning and big data. It has been shown that other approaches can also generalize well over smaller data sets, for example SVMs, which are used in this study. This is applied to many areas of science, for example chemistry or medicine, where data sets used in machine learning algorithms can have a similar size to the one used here (e.g., Haynes & Rees, 2005).

3. In my opinion the color specificity of emotion and again country-specificity is culturally determined and requires analysis of knowledge available in languages. It also requires for taking context into consideration, say for example, RED for a soldier at

war field, RED for a person driving vehicle, or RED in the flag of a country, can not be related to the same emotion. Hence, this paper requires more debate and raises many unanswered questions.

- We fully agree with both of your suggestions. First, we also think that the (relatively small) differences we observed between the four countries are likely due to cultural differences, as we now discuss for the examples of WHITE in China and PURPLE in Greece. However, additional research is needed to rule out alternative explanations. As stated above, the focus of the current paper is on the machine learning measures. For this reason, we do not provide a full discussion of potential cultural origins of the differences between countries, but plan to do so in other (upcoming) papers. We would also like to note that we had expected more pronounced differences between countries. When we planned this study, asking colleagues and friends, most of them would guess a much stronger differences between cultures and languages. Yet, our findings show that color-emotion associations show some variation between countries, but still are relatively similar. This observation is thus even more important. Second, color can of course have different meanings depending on the context and humans show different color preferences depending on the context (Jonaskaite et al., 2016; Schloss, Strauss, & Palmer, 2013). Still, there are some meanings of color that generalize across different contexts. For instance, in most countries RED is an indication of danger, in different context such as a person's face turning red (due to anger), a red traffic light, or a red warning signal on a control panel. Thus, it remains for future research to investigate whether changes in context would produce significant changes in color-emotion associations.

References

- Haynes, J. D., & Rees, G. (2005). Predicting the orientation of invisible stimuli from activity in human primary visual cortex. *Nature Neuroscience*, 8(5), 686-691. doi: 10.1038/nn1445
- Jonaskaite, D., Mohr, C., Antonietti, J. P., Spiers, P. M., Althaus, B., Anil, S., & Dael, N. (2016). Most and least preferred colours differ according to object context: New insights from an unrestricted colour range. *Plos One*, 11(3). doi: 10.1371/journal.pone.0152194
- Schloss, K. B., Strauss, E. D., & Palmer, S. E. (2013). Object color preferences. *Color Research and Application*, 38(6), 393-411. doi: 10.1002/col.21756

Dr. Daniel Oberfeld, Ph.D.
Associate Professor

Johannes Gutenberg-Universität Mainz

Wallstrasse 3
Room 06-416
D-55122 Mainz
Germany

Tel. +49 6131 39-39274
Fax +49 6131 39-39268

25. March 2019

Appeal against the negative decision concerning our revised manuscript "A machine learning approach to quantifying the specificity of color-emotion associations and their cultural differences" (RSOS-181134.R1)

oberfeld@uni-mainz.de

<http://www.staff.uni-mainz.de/oberfeld/>

Dear Editor,

with the current letter, we would like to appeal against your decision to reject our revised version of our manuscript entitled "A machine learning approach to quantifying the specificity of color-emotion associations and their cultural differences".

We had submitted a revised version of our original submission, together with a response letter addressing all of the comments of the reviewers. After a delay of several months, we received only a comment by one single reviewer, who did not at all specify which points he or she thought were not adequately addressed by our revision, and did not provide a single scientific argument. For this reason, the negative decision appears to be based on the evaluation of the subject editor. Judging from the comments of the editor, we think that the decision concerning our paper is based on **three misunderstandings or misconceptions**, which we address in the following. We had already addressed all of the issues raised by the editor in the response letter we submitted together with the revision. Here, we are focusing on the points noted by the editor, and explain our arguments in greater depth than in the previous response letter.

1) Sample size. The reviewers indicated that they consider our dataset to be relatively small. Our data consist in color-emotion ratings from 711 participants collected in four countries. The sample size is certainly small when having big data scenarios with thousands if not millions of participants in mind ("big data", "data mining" context). We suggest that these big data scenarios are very different to our current situation. Here, we performed a controlled experiment. We were sure participants had understood the task (we had included a manipulation check). We had a pre-conceived structure and assumption about the data (colour and emotions are associated, we preselected colours and emotions, including the way we present them in an experimental procedure). We did not use a data mining approach where structure was inferred through algorithms. Within this context, we argue that our sample size can be considered medium to large in size. We do so accounting for relevant literature in our respective field, i.e., previous psychological studies on cross-cultural color-emotion associations. For instance, Hupka, Zaleski, Otto, Reidl, and Tarabrina (1997) report data from 660 participants (in 5 countries), while Madden, Hewett, and Roth (2000) reported data from 253 participants (in 8 countries), all per-country sample sizes were smaller than 50. With more than 100 participants per country, we suggest that our data

represent a meaningful sample of experimentally gathered color-emotion associations in the four countries. Also, in contrast to several previous investigations, our sample is not limited to young adults.

Another point regarding sample size is that the dimensionality of our data set in relation to the sample size is comparable to other applications of machine learning in psychology, chemistry, and the neurosciences. For instance, in an influential study from the neurosciences (published in *Nature Neuroscience*), a classifier was trained to predict the visual stimulus an observer was looking at from the activity of 100 fMRI voxels, using only 238 training volume acquisitions and 34 test volume acquisitions (Experiment 1; Haynes & Rees, 2005).

2) **Classifier accuracy.** The reviewers had commented that they consider our classifier accuracies to be relatively low. In the response letter accompanying the first revision, we explained that in our study, the intention of training a classifier was (and still is) not to present a classifier of a very high accuracy, as it is typical in the machine learning literature applied to, e.g., the detection of junk mail or automatic speech recognition. Rather, we use the empirically obtained classifier accuracy as a **metric** that reveals the structure of color-emotion associations. As discussed in the paper, an AUC of 0.83 for the color classification, which is significantly higher than performance by random guessing (backed up by the results of the randomization tests), tells us that there is a "signal" in the data and we can pick it up. For our psychological research question, this indicates that the color-emotion associations are not arbitrary and idiosyncratic, but rather follow a systematic pattern across the 711 participants from four countries. At the same time, the classifier accuracies were, indeed, much lower than 100%. This fact speaks against a one-to-one relationship in color-emotion associations. This observation is informative for the psychological community, because such one-to-one relationships have been claimed in the literature. Our results indicate that the same emotion is associated with several different colors, and the patterns of associated emotions vary between subjects, as described in the paper. While training a superior classifier algorithm (e.g., deep learning) on a much larger data set might somewhat improve the classifier accuracy, we are convinced that the strongest limitation is in the data, due to the fact that **psychologically** a given color is not consistently associated with only very a specific set of emotions. Thus, the limited accuracy is not a flaw but represents an important psychological finding concerning color-emotion associations, as we discuss in the paper.

Put differently, our intention is not to present a classifier that can predict the evaluated color from the 20 emotion associations at an accuracy of, say, 99% (like for instance a junk email classifier), but we rather propose the classifier accuracy as an **empirical metric** for the specificity of color-emotion associations. That is what we meant when using the term "exploratory" in our response letter accompanying the revised version. The term "exploratory" was seemingly an unfavorable term for what we wanted to express. We have the impression that this term seemed to have caused further misunderstandings, by both the reviewer and the subject editor.

We hope we could clarify the value of the classifier accuracies to our study question. Here, we would like to reiterate the novelty of the question and approach using machine-learning to quantify the specificity of color-emotion associations and their cultural difference. We provide to the wider community a new empirical approach when facing complex data such as the current set.

3) **Robustness of the machine learning analyses.** The reviewers and the editor wondered how robust our classifier accuracies are. This point relates to the previous two points (sample size, classifier performance). We, of course, agree (and stated so in the paper and in the previous response letter) that a much larger sample size would, by definition, provide more representative data. A larger sample would be advantageous for training the classifiers (see also the points above). It is important to note, however, that we measured the classifier accuracies using 10-fold cross-validation. In this approach, the classifier is repeatedly trained on a random subset of the data (training set) and then the accuracy is measured for the subset of the data (test set) not included in the training set. The machine-learning literature shows that this is a powerful approach for avoiding problems with overfitting. If the classifier had picked up random or idiosyncratic patterns present only in the training data sets, the classifier accuracy would have been much lower across the 10 cross-validation runs.

In addition, the editor commented that a test on an independent sample was missing. Between submitting the first version of the current manuscript and today, we have complete data from 30 countries (same inclusion and exclusion criteria as described here). The report on the data from these 30 countries (including the current four countries) is ready for submission for a publication elsewhere. The report on this larger sample focusses on the cross-cultural comparison. To account for the editor's

concern, we applied the color classifier trained on the data from China, Greece, Germany, and the UK to the data from the remaining 26 countries. This 26-countries data set contained ratings of color-emotion association from 3222 participants. The classification accuracy was 0.304 and the AUC was 0.704. While these values are somewhat lower than for the four-countries data set (38.7% correctly classified instances, AUC = 0.83), these values are well above chance level. We hope that this observation assures the editor regarding the robustness of our results.

Put differently, we successfully tested the machine learning algorithm on an independent sample, and obtained comparable accuracy values. This is no surprise, given that the classifier accuracy for the 4-countries data set was measured using cross-validation. Taken together, these results show that there is no need to worry about the robustness of our results.

In addition to these explications so far, we want to emphasize that our manuscript makes two important contributions. First, it is the first study ever that obtained ratings for associations between colors and 20 discrete emotions in a relatively large, international sample ($N = 711$). Thus, the "raw" data already represent an important contribution to the literature on color and emotion. The field is increasing and the current approach is likely welcomed by others. Second, the focus of the paper is on the proposed machine-learning measure to estimate the specificity of color-emotion associations and their cultural differences. This type of analyses has not previously been used in research on emotion. For this reason, our paper makes an important methodological contribution that is highly likely being used in future research.

Based on our arguments above, we suggest that the rejection of our paper was unjustified. We would like to ask the editors to reconsider their decision. In the unlikely event that additional reviews are required, we suggest to ask the following experts:

John-Dylan Haynes, Bernstein Zentrum Berlin, sekretariat.haynes@bccn-berlin.de (application of machine learning techniques to neuroscience data)

Karen Schloss, University of Wisconsin–Madison, kschloss@wisc.edu (color preferences and color-meaning associations)

Alice Skelton, University of Sussex, UK, A.E.Skelton@sussex.ac.uk (colour categorisation)

Adam Pazda, University of South Carolina Aiken, USA adamp@usca.edu (colour and approach/avoidance motivation)

Jennifer Fugate, University of Massachusetts Dartmouth, USA, jfugate@umassd.edu (colour and emotion, emotion categorisation)

Best regards

Daniel Oberfeld (on behalf of all authors)

References

- Haynes, J. D., & Rees, G. (2005). Predicting the orientation of invisible stimuli from activity in human primary visual cortex. *Nature Neuroscience*, 8(5), 686-691. doi: 10.1038/nn1445
- Hupka, R. B., Zaleski, Z., Otto, J., Reidl, L., & Tarabrina, N. V. (1997). The colors of anger, envy, fear, and jealousy: A cross-cultural study. *Journal of Cross-Cultural Psychology*, 28(2), 156-171. doi: 10.1177/0022022197282002
- Madden, T. J., Hewett, K., & Roth, M. S. (2000). Managing images in different cultures: A cross-national study of color meanings and preferences. *Journal of International Marketing*, 8(4), 90-107. doi: 10.1509/jimk.8.4.90.19795

Dr Narayanan Srinivasan
Royal Society Open Science

Department of Psychology
Section Experimental Psychology

Dr. Daniel Oberfeld, Ph.D.
Associate Professor

Johannes Gutenberg-Universität Mainz

Wallstrasse 3
Room 06-416
D-55122 Mainz
Germany

Tel. +49 6131 39-39274
Fax +49 6131 39-39268

8. July 2019

Revised manuscript "A machine learning approach to quantifying the specificity of color-emotion associations and their cultural differences" (RSOS-190741)

oberfeld@uni-mainz.de

<http://www.staff.uni-mainz.de/oberfeld/>

Dear Editor,

We are grateful to you and the reviewers for the positive evaluation of our manuscript and for the helpful comments.

We are now submitting a revised version for which we carefully considered all comments and implemented the changes where appropriate. We marked the edits in blue ink in the revised manuscript.

Our answers to the comments in the decision letter are listed below. We hope that the changes have made the paper acceptable for publication in your journal.

Best regards,

Daniel Oberfeld (on behalf of all authors)

Response (indicated by bullets and blue ink) to the comments of Reviewer 3

Comments to the Author(s)

Apologies, I missed that this was a resubmission with my initial review. I have looked at their response (I don't have access to the original reviews) and will discuss the three points that they discuss here.

- Complaints about sample size- I agree with the authors that their sample size of 711 participants is more than adequate in this context. I also agree that in comparison to a large 'big data' set, 711 is 'small'. However, the current sample size is bigger than is often found in many studies of color-emotion (or more generally in studies of color cognition), and so makes an important contribution to the field. Having looked at the task (via the link which the authors make available in text) I think it is appropriate for the research questions being asked, and their screening of participant data allows for confidence in the responses that they acquired.

- Classifier accuracy - The unique contribution of this study is that it uses an established method in a novel way. In many parts of color cognition, researchers are rarely looking for a single variable which can account for all the variance in their data.

Consider for example the field of color preference - there are many different variables which have

been shown to influence color preference, just some of which are listed below:- ease of naming of colors and prototypicality of color; ecological valence theory (that objects are liked as a result of the interaction between either experience over the lifetime with objects of that color, or potentially over the course of evolution); color preference in children has been tied to developing an identity of the self (pink for girls, blue for boys); relationship with the S/(L+M) and L/(L+M) pathways of color vision.

As a result, most models of color preference account for a proportion of variance within data (and vice versa, would *predict* a proportion of the variance) - no single model would expect to account for all the variance in a dataset, although clearly some will account for more than others (say, 70% vs 30%). This *amount* that can be explained is informative in understanding the hierarchy of these variables in a color perception task. The same is true in studies of color-emotion. The valuable insight gained from the data analysis in the current paper is that there is a metric which will allow for quantification of the strength of different variables impact on color-emotion associations.

Robustness of the model : I agree with the points the authors raise in their response - if the concern here is related to the previous two points (sample size and accuracy), then I think concerns about inaccuracy aren't correct. There is always, as with all cognitive based models, some room for error - the authors make this clear in their manuscript. The extra data mentioned, from more countries, adds to the authors point here.

- We thank the reviewer for the positive evaluation of our paper.

** Initial review submitted 01/05/19 - these suggested changes are very minor, and I think the paper could be accepted without significant, or possibly any, changes to these points. Apologies to the authors- had I realised the initial comments I would have focused my review on those points.

This was a really interesting paper providing interesting insights using a novel method into color-emotion associations. The data itself is very rich, providing both qualitative and quantitative insights into the relationship of color-emotion, and there is a lot of ground covered in terms of scope and literature.

My comments are very minor. I expect that that these changes would be at most a handful of sentences, but would strengthen some of the arguments put forward in the discussion in particular -

1. Discussion - line 498 there is some discussion about perceptual similarity. It's a bit unclear what is really meant by perceptual similarity here. I'd be cautious, for example, about using the phrase "orange being a mixture of red and yellow" - there are models of color which are not additive in this way after all, where orange could be described in terms of e.g. cone excitation, and so independent of the red and yellow hues, but still perceptually similar. This can be rectified by expanding very slightly (by maybe one sentence) on what perceptual similarity means here, or just removing the phrase above.

- We agree with the reviewer that "perceptual similarity" of orange and yellow depends on the color model. Thus, we adapted the sentence in the revised manuscript as follows: "*Orange and yellow are perceptually similar colors, with similar hue angles in for instance the CIELAB color space [54], and both are being considered as warm hues [32].*"

2. Discussion - around 507. There's a great discussion of the relationship between focal color terms and the relationships reported in the paper. I agree with the authors about the convergence of focal terms in the color space, but do have a remaining question about the color terms they've used. For example, Greek - one of the languages chosen, I believe has two basic color terms for blue: two BCTs for blue, μπλε [blé] "blue" and γαλάζιο [yalazjo] "light blue", meaning all of English BCTs are covered in the study, but not all Greek BCTs. Any differences between BCTs in these languages should be acknowledged at this point in the discussion. It doesn't threaten any of their findings - the authors are clear on the variability in e.g. focal terms across participants and languages, but it may be helpful to flag this to readers. One or two sentences at most.

- Thank you for this interesting and relevant comment. After a little bit of research and discussion with our Greek co-author (MPP), we agree regarding the existence of the 12th basic color term in the Modern Greek language. We have included this consideration in the revised manuscript: *“This could especially be the case for Greek speakers who have an additional 12th basic color term for LIGHT BLUE (“γαλάζιο”) [57]. In the current study, the Greek term for LIGHT BLUE was assessed in the TURQUOISE category. Thus, the Greek participants might have pictured a darker shade of blue when given the color term BLUE (“μπλε”) than participants from the other countries.”*

3. Discussion- around 559. Although the authors do discuss cultural differences in use of color - it still doesn't tell us about how exactly these associations are learned and the mechanisms behind it. e.g. is it semantic mediation? I appreciate the results of this study can't tell us this, but a single sentence or reference to this would help the reader put this research into context.

- In the revised manuscript, we have added some suggestions for the origins of color-emotion associations: *“The origins of these cross-cultural differences in color-emotion associations cannot be determined from the current study. Nevertheless, some cross-cultural differences could potentially be explained through the existence of different color metaphors (conceptual metaphor theory, Lakoff & Johnson, 1999; Soriano & Valenzuela, 2009), which would be semantically mediated, or different perceptual cross-modal associations between colors and emotional stimuli (statistical correspondence framework; Spence, 2011).”*

Reviewer: 4

Comments to the Author(s)

When I accepted the review, I was not aware that this manuscript had been previously reviewed. I was therefore surprised to see a rebuttal letter from the authors. This letter addressed most of my more serious potential concerns very well. I will therefore only focus on additional issues.

The topic is interesting and warrants more research. This paper seems a good first step in that direction.

- We are glad to read that most of the more serious potential concerns have already been addressed in the response letter and that the reviewer is positive about the current manuscript.

The introduction is engagingly written, but somewhat lacking in rationale. For example, the authors propose an ingroup advantage in color-decoding accuracy but (a) they do not test color decoding accuracy, but rather color-emotion associations, (b) the proposed reference (31) has little to say about ingroup advantages and more importantly does not refer to color at all. If the authors want to use this concept in some sort of metaphorical manner, they need to explain why they think this to be helpful.

- At this point, it is important to distinguish between a) the task of the participants (rate the associations between color terms and discrete emotions) and b) our proposed machine-learning approach for analyzing these empirical data. In one of the analyses, the classification algorithm was indeed trained to decode (identify) the color term a participant evaluated from the pattern of 20 emotion association ratings. In this context, it is a meaningful definition of an in-group advantage to compare the color-decoding accuracy of a classifier trained on data from the same or from a different country. If the color-emotion associations were identical between the four countries, then the accuracy in decoding the evaluated color from the 20 ratings in, e.g., the German sample should not differ between a) a classifier trained on data from Germany and b) a classifier trained on data from, e.g., the UK. If, however, there are systematic differences between the color-emotion associations in the four countries, then an in-group advantage should be observed, in the sense that the classifier accuracy for decoding the color in the German sample is higher when the classifier was trained on the German data rather than on the data from a different country. We compare the in-group advantage of our classifiers (not of the participants!) to the in-group advantage shown by humans when decoding discrete emotions from facial expression, because we feel that such a comparison concerning cultural differences in the processing of emotion is interesting.

I understand that given the authors' goals sample size can be considered adequate. However, the samples are heavily based towards women. Given that stereotypical color preferences differ between the sexes I think this poses a problem in terms of interpretation.

- We understand the reviewer's concern regarding the gender composition of the current samples, and already address this issue in the discussion, also noting that precisely the machine learning approach we propose here for quantifying differences between countries could also be used for quantifying differences in the color-emotion associations between different genders. That being said, we are of course well aware of the gender differences in color preferences (our recent publication treated this question in children and adults; Jonauskaitė et al., 2019, *Sex Roles*). However, several studies suggest that color preferences are different from color-emotion associations. For instance, yellow seems to be an unpleasant and disliked color (e.g., Jonauskaitė et al., 2019; Palmer & Schloss, 2010; Valdez & Mehrabian, 1994). However, it is consistently associated with joy (see the current study, and, e.g., Burkitt & Sheppard, 2014; Dael, Perseguers, Marchand, Antonietti, & Mohr, 2015; Kaya & Epps, 2004; Lindborg & Friberg, 2015). In line with these results suggesting a dissociation between color preferences and color-emotion associations, color-emotion associations collected by us using the same methods as in the present paper but in a much larger sample (4,598 participants in 30 countries; manuscript in preparation), shows a similarity of 98.7% between the color-emotion associations in men and in women.

Why were the color terms translated? This is an online study – why not show the colors?

- The answer to the second question is very simple: it is technically impossible to control the display color in an online study, due to the large variation in displays (think smart phones, different computer monitors etc.) and ambient lighting conditions. In other words, it is not possible to present a colorimetrically defined color outside a controlled and properly calibrated laboratory setting. For this reason, we used color terms, as already mentioned in the discussion (*"We used linguistic color terms to assess semantic associations as it is impossible to control the colorimetric coordinates of colors in an online survey."*). We have now added a similar statement to the methods section. In the discussion, we already address some potential implications of using color terms rather than color patches. Importantly, we state that it remains to be shown how color-specific and culture-specific emotion associations are across a wide range of visually presented colors varying in hue, saturation, and lightness, and that the machine learning-based measures developed here can be applied to such data. On a more general level, we agree that it is an interesting question whether using linguistic color stimuli (i.e., color terms) or perceptual color stimuli (i.e., color patches) should be used in color-emotion research. Previously, there have been numerous studies that used color terms (Adams & Osgood, 1973; Hupka et al., 1997; Soriano & Valenzuela, 2009; Sutton & Altarriba, 2016) or color patches (Allen & Guilford, 1936; Fugate & Franco, 2019; Hanada, 2018; Kaya & Epps, 2004; Valdez & Mehrabian, 1994; Wilms & Oberfeld, 2017) to test color-emotion relationships.
- Concerning the first question, the reason why we translated color terms was because we wanted participants to complete the survey in their native language. Native language can influence the way colors are perceived and categorized (e.g., Maier, M., & Abdel Rahman, R. (2018). Native language promotes access to visual consciousness. *Psychological Science*, 29(11), 1757–1772. <https://doi.org/10.1177/0956797618782181>). We believe that testing only native speakers of an official language in each country is a strong point of the current study. It allows for a cross-country comparison without the uncontrolled influence of language.

Participants had to make 240 fairly demanding decisions in an online study. The authors do not report any attempt to check for attention. I would expect participants to fatigue and start giving more random answers. Also, in online studies it is never clear what else participants are doing at the same time. Attention checks are typically used and generally suggested for this type of study, especially when the task is long and tedious.

- We understand that having to provide 240 ratings may appear demanding at first sight. However, participants saw only twelve webpages, each time with one color term and the Geneva Emotion Wheel (GEW) showing 20 discrete emotions below it. On each of these 12 pages, participants made 20 decisions by selecting the degree of association between the emotions and the color term on the GEW. The rating "0" was automatically assigned to all the emotions not chosen by the participant. We

chose the Geneva Emotion Wheel because it provides a user interface with a quick and efficient rating (e.g. Tran, 2004; Caicedo & van Beuzekom, 2006). In fact, participants took on average only 13 minutes to complete the survey (line 256). We believe that this relatively small amount of processing time is acceptable for a research study and did not cause extensive fatigue. At the same time, as discussed in the methods section, we made some checks for proper engagement in the task by excluding participants who were very quick (took < 5 min on the main task) or very slow (took > 60 min) in completing the survey, or did not show minimal engagement with the task (i.e., spent < 20 s on the first four color terms).

The discussion is very long and verbose and much of it repeats the results section. There is even a results figure in the discussion. What is missing is a more solid theoretical integration.

Also, while I understand that the authors consider their approach a first step with the goal to show that there is a consistent signal in color-emotion associations, I was missing an idea where to go from here.

- We would like to add that a clear theoretical framework is still missing in color-emotion studies, and a consequence of this is that the current study is not intended to test a specific theory but to first establish large-scale empirical data concerning color-emotion associations and a new method for analyzing them, which we hope will be useful for formulating improved models/theories in the future. Still, as mentioned above, following the request of Reviewer 3 we have implemented some suggestions on the theoretical frameworks that would be helpful in explaining color-emotion associations (see our response to point 3 made by Reviewer 3). We hope this addition goes in line with your suggestion to provide more theoretical integration.